# Hybrid Regularization Improves Diffusion-based Inverse Problem Solving

**Hongkun Dou**[1], **Zeyu Li**[1], **Jinyang Du**[1,3], **Lijun Yang**[1], **Wen Yao**[2,*] **& Yue Deng**[1,3,*]
[1] Beihang University    [2] Chinese Academy of Military Science
[3] Beijing Zhongguancun Academy
`{douhk,lizeyu123478,dujinyang,yanglijun,ydeng}@buaa.edu.cn,`
`wendy0782@126.com`

## Abstract

Diffusion models, recognized for their effectiveness as generative priors, have become essential tools for addressing a wide range of visual challenges. Recently, there has been a surge of interest in leveraging Denoising processes for Regularization (DR) to solve inverse problems. However, existing methods often face issues such as mode collapse, which results in excessive smoothing and diminished diversity. In this study, we perform a comprehensive analysis to pinpoint the root causes of gradient inaccuracies inherent in DR. Drawing on insights from diffusion model distillation, we propose a novel approach called Consistency Regularization (CR), which provides stabilized gradients without the need for ODE simulations. Building on this, we introduce Hybrid Regularization (HR), a unified framework that combines the strengths of both DR and CR, harnessing their synergistic potential. Our approach proves to be effective across a broad spectrum of inverse problems, encompassing both linear and nonlinear scenarios, as well as various measurement noise statistics. Experimental evaluations on benchmark datasets, including FFHQ and ImageNet, demonstrate that our proposed framework not only achieves highly competitive results compared to state-of-the-art methods but also offers significant reductions in wall-clock time and memory consumption. Code is available at https://github.com/deng-ai-lab/HRDIS.

## 1 Introduction

Diffusion models (Ho et al., 2020; Song et al., 2020b), celebrated for their ability to model complex high-dimensional distributions, are increasingly applied across a broad spectrum of fields. These applications span from image generation (Dhariwal & Nichol, 2021; Saharia et al., 2022b; Rombach et al., 2022), and video synthesis (Cho et al., 2024; Ho et al., 2022; Rühling Cachay et al., 2024) to molecular design (Guan et al., 2023; Hoogeboom et al., 2022; Xu et al., 2022), and protein generation (Wu et al., 2024a; Trippe et al., 2022; Watson et al., 2023). The wealth of knowledge accumulated within pre-trained diffusion models, derived from vast datasets, empowers their utilization as powerful prior models. This capability is particularly valuable for reconstructing accurate outputs from incomplete measurement signals.

Recent research has highlighted the potential of diffusion models as general-purpose posterior samplers in solving inverse problems. For instance, Kawar et al. (2022) pioneered DDRM, innovatively integrating the observation into the inverse denoising process within the spectral domain through singular value decomposition. In a similar vein, Wang et al. (2022) introduced DDNM, which methodically adjusts the inverse diffusion process to align with the observation's null space. Despite their novelty, these approaches are limited to addressing linear degenerations and often result in low-fidelity images in practical applications. To improve upon these limitations, subsequent innovations like DPS (Chung et al., 2022) and ΠGDM (Song et al., 2022) have been developed, focusing on approximating the score of posterior distributions to achieve image restoration. However, their dependence on an unimodal estimation for clean samples has led to issues with precision. Additionally, the requirement

---

*Correspondence to W. Yao and Y. Deng

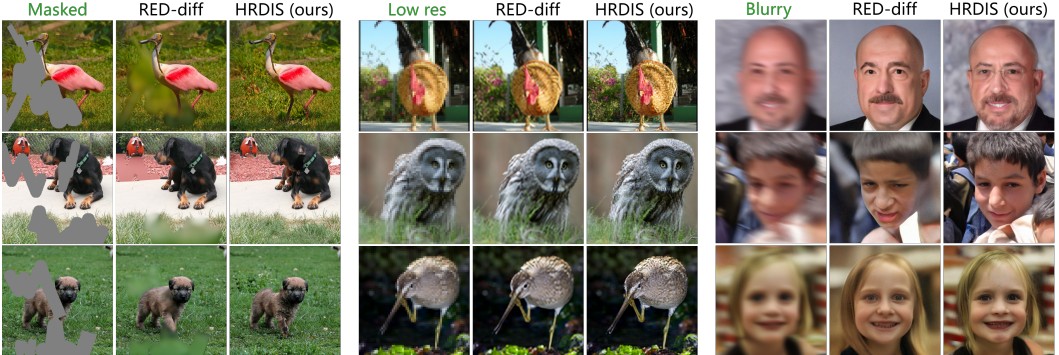

Figure 1: Comparison between RED-diff (Mardani et al., 2024) and our proposed HRDIS for image inpainting (left), super-resolution (middle) and nonlinear deblurring (right) tasks. RED-diff adopts the denoising process for regularization, which frequently leads to blurry and detail-lacking reconstructions, particularly when addressing intricate image inversion challenges. In contrast, HRDIS consistently outperforms RED-diff, showcasing notably improved results.

to compute the Jacobian matrix of the score network with DPS and ΠGDM introduces significant computational demands and potential instability, posing challenges for efficient implementation.

RED-diff (Mardani et al., 2024) provides a novel perspective to addressing diffusion-based inverse problems, conceptualizing them within a variational objective framework. This framework employs a denoising process as a regularization technique to alleviate the ill-posedness of the inverse problem. Such an optimization-based approach circumvents the necessity for backpropagation through the score network, facilitating compatibility with existing efficient stochastic optimization methods. Despite these advantages, RED-diff encounters practical challenges, notably a mode-seek tendency that often leads inversion results to converge on feature-averaged estimation (see Figure 1).

In this study, we investigate the underlying issues of RED-diff, particularly optimization challenges caused by gradient inaccuracy, which we attribute to its reliance on single-step denoising estimation as a bootstrap target. Drawing upon principles for the distillation of diffusion models (Song et al., 2023b; Song & Dhariwal, 2023), we develop the Consistency Regularization (CR) technique. This approach enables more stable gradient provision, facilitating the regularization of the target signal for recovery, and obviates the need for computationally intensive simulation of ordinary differential equations (ODEs). Building on this foundation, we further introduce a comprehensive Hybrid Regularization (HR) framework. It provides a unification of RED-diff and CR, amalgamating their benefits to enhance the performance of inverse problem solving markedly.

We introduce our framework, **H**ybrid **R**egularization for **D**iffusion-based **I**nverse problem **S**olving (HRDIS), which serves as a versatile solution for addressing a wide range of inverse problems. we evaluate HRDIS across various tasks, encompassing both linear (inpainting, super-resolution, and compressed sensing) and nonlinear (phase retrieval, high dynamic range, and nonlinear deblurring) scenarios, using two widely recognized benchmark datasets: FFHQ (Karras et al., 2019) and ImageNet (Deng et al., 2009). Additionally, we demonstrate the framework's adaptability to various noise types. Our results show that HRDIS achieves performance that matches or exceeds state-of-the-art methods, while significantly reducing wall-clock time and memory usage.

## 2 BACKGROUND

### 2.1 SCORE-BASED DIFFUSION MODELS

The diffusion model (Ho et al., 2020; Song et al., 2020b) introduces a forward stochastic differential equation (SDE), denoted as $\{\mathbf{x}_t\}_{t \in [0,1]}$, which gradually perturbs the initial data $\mathbf{x}_0 \sim p_{\text{data}}$ into noise:

$$\mathrm{d}\mathbf{x} = \mathbf{f}(\mathbf{x}, t)\mathrm{d}t + g(t)\mathrm{d}\mathbf{w}, \qquad (1)$$

where $\mathbf{f}(\cdot, t)$ and $g(t)$ represent the drift and diffusion coefficients, respectively, and $\mathbf{w}$ is the standard Wiener process. We denote the marginal probability densities w.r.t $\mathbf{x}_t$ as $p_t(\cdot)$. By selecting appropriate coefficients, we can reparameterize $\mathbf{x}_t = \alpha_t \mathbf{x}_0 + \sigma_t \boldsymbol{\epsilon}$, where $\boldsymbol{\epsilon} \sim \mathcal{N}(\mathbf{0}, \boldsymbol{I})$ and obtain a standard

Gaussian density at $t = 1$. The forward SDE above is coupled with a probability flow (PF) ODE, which can be expressed as:

$$d\mathbf{x} = \left[\mathbf{f}(\mathbf{x}, t) - \tfrac{1}{2}g^2(t)\nabla_{\mathbf{x}} \log p_t(\mathbf{x})\right] dt, \tag{2}$$

The above equation introduces a time-dependent score function $\nabla_{\mathbf{x}} \log p_t(\mathbf{x})$, which can be approximated minimizing denoising score matching (Hyvärinen & Dayan, 2005) objectives using a neural network $\boldsymbol{\epsilon}_\theta$,

$$\min_\theta \mathbb{E}_{t, \mathbf{x}_0 \sim p_{\text{data}}, \boldsymbol{\epsilon} \sim \mathcal{N}(\mathbf{0}, \boldsymbol{I})} \|\boldsymbol{\epsilon}_\theta(\alpha_t \mathbf{x}_0 + \sigma_t \boldsymbol{\epsilon}, t) - \boldsymbol{\epsilon}\|_2^2. \tag{3}$$

After training, we can approximate $\nabla_{\mathbf{x}} \log p_t(\mathbf{x}) \approx -\boldsymbol{\epsilon}_\theta(\mathbf{x}, t)/\sigma_t$ and simulate such ODE in reverse time to sample from the underlying distribution $p_{\text{data}}$.

Given that simulating the ODE can be computationally intensive, often requiring hundreds of iterations, consistency distillation (CD) (Song et al., 2023b) has recently been introduced to learn a direct mapping from noise to data. This mapping is parameterized as $f_\phi(\cdot, \cdot)$ and is trained to maintain self-consistency:

$$\min_\phi \|f_\phi(\mathbf{x}_{t+\varepsilon}, t + \varepsilon) - \text{sg}(f_\phi(\mathbf{x}_t, t))\|_2^2 \tag{4}$$

where $\varepsilon > 0$, and $\text{sg}(\cdot)$ denotes the stop-gradient operator. Here, $\mathbf{x}_{t+\varepsilon}$ and $\mathbf{x}_t$ are two adjacent points on the same PF ODE trajectory. As the model easily learns the cases for small $t$, this loss function propagates the endpoint of the trajectory toward $t = 1$, promoting a one-step approximation of the ODE solution.

## 2.2 DIFFUSION MODELS FOR INVERSE PROBLEMS

The inverse problem arises in various applications across diverse domains. Formally, the general model for the inverse problem can be expressed as:

$$\mathbf{y} = \mathcal{A}(\mathbf{x}_0) + \boldsymbol{\eta}, \quad \mathbf{y}, \boldsymbol{\eta} \in \mathbb{R}^n, \mathbf{x}_0 \in \mathbb{R}^m, \tag{5}$$

where $\mathcal{A}(\cdot) : \mathbb{R}^m \to \mathbb{R}^n$ is the measurement operator and $\boldsymbol{\eta}$ is the noise in the measurement process. When it is Gaussian noise, $\boldsymbol{\eta} \sim \mathcal{N}(0, \gamma^2 \boldsymbol{I})$, then we have $p(\mathbf{y}|\mathbf{x}_0) \sim \mathcal{N}(\mathcal{A}(\mathbf{x}_0), \gamma^2 \boldsymbol{I})$. In practical applications, the collection of measurements often results in degradation relative to the original signal. Notably, when $n < m$, the problem becomes ill-posed, necessitating the incorporation of a regularizer or prior for deriving a meaningful solution. Recent research has demonstrated that diffusion models can serve as plug-and-play generative priors for sampling from posterior distributions, thereby obviating the need for extensive fine-tuning for specific tasks. To address this challenge, several classes of methods have been proposed. One widely used approach is the **projection-based methods** (Song et al., 2020b; Kawar et al., 2022; Wang et al., 2022), which constrains the reverse-time denoising process to the subspace of measurements. However, these heuristics often fail to harmonize the generated samples with the known region (Lugmayr et al., 2022).

Alternatively, **guidance-based methods** have been proposed (Chung et al., 2022; Song et al., 2022) . The gradient of the log-likelihood, i.e. $\nabla_{\mathbf{x}_t} \log p(\mathbf{y}|\mathbf{x}_t)$, is approximated and integrated into Eq. 2 for posterior sampling from the measurement $\mathbf{y}$. Specifically, in order to obtain $p(\mathbf{y}|\mathbf{x}_t) = \int p(\mathbf{y}|\mathbf{x}_0)p(\mathbf{x}_0|\mathbf{x}_t)d\mathbf{x}_0$, DPS (Chung et al., 2022) approximate $p(\mathbf{x}_0|\mathbf{x}_t)$ with an Dirac delta distribution, i.e. $p(\mathbf{x}_0|\mathbf{x}_t) \approx \delta(\mathbf{x}_0 - \hat{\mathbf{x}}_{0|t})$, where $\hat{\mathbf{x}}_{0|t} \triangleq \mathbb{E}[\mathbf{x}_0|\mathbf{x}_t] = (\mathbf{x}_t - \sigma_t \boldsymbol{\epsilon}_\theta(\mathbf{x}_t, t))/\alpha_t$ represents the single-step denoising estimation by Tweedie's formula (Robbins, 1992). Thus, we have the following approximation,

$$\nabla_{\mathbf{x}_t} \log p(\mathbf{y}|\mathbf{x}_t) \approx -\tfrac{1}{\gamma^2} \nabla_{\mathbf{x}_t} \|\mathbf{y} - \mathcal{A}(\hat{\mathbf{x}}_{0|t})\|_2^2. \tag{6}$$

While these methods prove effective and typically produce clear reconstruction results, they are not exempt from limitations. In particular, the gradient backpropagation process of the network $\boldsymbol{\epsilon}_\theta$ is computationally expensive and prone to instability.

Recently, Mardani et al. (2024) introduced an **optimization-based method**, RED-diff, with promising results. They tackle the problem by treating posterior sampling as a variational optimization task,

$$\min_q D_{KL}(q(\mathbf{x}_0|\mathbf{y})\|p(\mathbf{x}_0|\mathbf{y})), \tag{7}$$

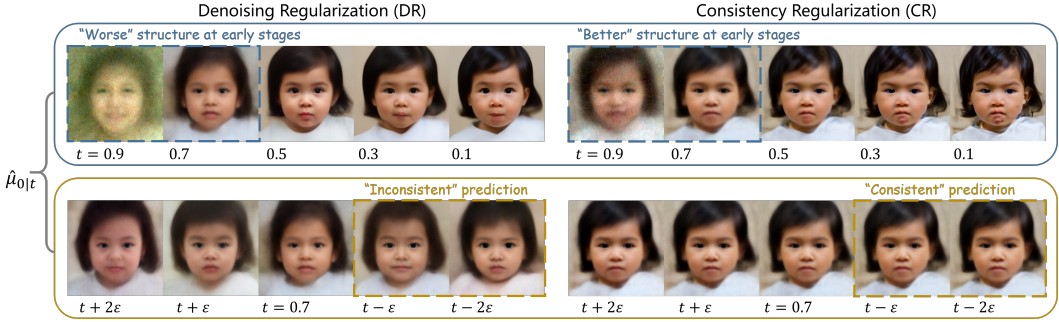

Figure 2: Comparison of intermediate denoised images $\hat{\mu}_{0|t}$ between DR and CR (Sec. 3.2). The top row displays an equally spaced visualization as $t$ progresses from 1 to 0. The bottom row presents $\hat{\mu}_{0|t}$ across 5 consecutive timesteps. CR demonstrates more consistent generation and captures intricate features at earlier stages than DR. The task is shown 16× super-resolution.

where $q \triangleq \mathcal{N}(\mu, \rho^2 \boldsymbol{I})$ is the variational distribution. The aforementioned variational objective can be further simplified as:

$$\min_{\mu} \underbrace{\|\mathbf{y} - \mathcal{A}(\mu)\|_2^2}_{\text{reconstruction}} + \underbrace{\mathbb{E}_{t,\boldsymbol{\epsilon}}[\rho_t \|\boldsymbol{\epsilon}_\theta(\mu_t, t) - \boldsymbol{\epsilon}\|_2^2]}_{\text{regularization}}. \tag{8}$$

where $\mu_t = \alpha_t \mu + \sigma_t \boldsymbol{\epsilon}$ and $\rho_t$ is the weighting function that depends on the timestep. The first term serves as a reconstruction term, ensuring that the reconstruction results align with the observed data. Meanwhile, the second term resembles the score matching objective, serving as a form of regularization to alleviate the ill-posed nature of $\mu$. We designate this technique as Denoising Regularization (DR). Mardani et al. (2024) also demonstrate that the gradient of DR can bypass the need for the Jacobian matrix of $\boldsymbol{\epsilon}_\theta$. Consequently, the resulting gradient can be expressed as:

$$\nabla_\mu \mathcal{L}_{\text{DR}} = \nabla_\mu \|\mathbf{y} - \mathcal{A}(\mu)\|_2^2 + \mathbb{E}_{t,\boldsymbol{\epsilon}}[\omega_t(\boldsymbol{\epsilon}_\theta(\mu_t, t) - \boldsymbol{\epsilon})], \tag{9}$$

where Mardani et al. (2024) chooses $\omega_t = \sigma_t/\alpha_t$ to balance the two terms. At this juncture, we can integrate the above gradient into an off-the-shelf stochastic optimizer and utilize the optimized $\mu$ as the reconstructed signal. RED-diff boasts memory efficiency, circumventing the need for backpropagation within the neural network. Also, it offers the advantage of utilizing the degraded image as an initial point for optimization, eliminating the necessity of starting with Gaussian noise. However, it is notable that RED-diff encounters challenges related to mode collapse, resulting in blurry outcomes and limited diversity. Furthermore, the convergence of RED-diff typically requires thousands of optimization steps.

## 3 METHODS

In this section, we first address the practical challenges in Denoising Regularization (DR) (Sec. 3.1). Building on these findings, we introduce Consistency Regularization (CR) as an alternative approach to overcome the limitations of DR (Sec. 3.2). We then present a unified framework, termed Hybrid Regularization (HR), which seamlessly integrates DR and CR as complementary techniques. Through a comprehensive analysis, we demonstrate how HR effectively leverages the strengths of both DR and CR methodologies to achieve superior performance (Sec. 3.3).

### 3.1 ANALYSIS OF GRADIENTS IN DR

Mardani et al. (2024) point out that the gradient of the regularization term in Eq. 9 can also be interpreted as the difference between the predicted clean image and variable $\mu$. By performing algebraic manipulation, we derive it as follows:

$$\omega_t(\boldsymbol{\epsilon}_\theta(\mu_t, t) - \boldsymbol{\epsilon}) = (\mu_t - \sigma_t\boldsymbol{\epsilon})/\alpha_t - (\mu_t - \sigma_t\boldsymbol{\epsilon}_\theta(\mu_t, t))/\alpha_t$$
$$= \mu - \hat{\mu}_{0|t} = \nabla_\mu \|\mu - \text{sg}(\hat{\mu}_{0|t})\|_2^2, \tag{10}$$

where $\hat{\mu}_{0|t} \triangleq \mathbb{E}[\mu|\mu_t]$ represents the *single-step* denoising estimate at timestep $t$. This derivation clarifies the mechanism of Denoising Regularization (DR), with $\hat{\mu}_{0|t}$ serving as a bootstrapped

ground truth. Regularization is achieved by constraining $\mu$ to align with $\hat{\mu}_{0|t}$ during optimization, thus ensuring it remains within the distribution of clean data.

Despite the effectiveness of this approach, DR has notable limitations. Firstly, the bootstrapped ground truth $\hat{\mu}_{0|t}$ is highly sensitive to $\mu_t$, and the stochasticity of noise perturbations introduces significant uncertainty, as illustrated in Figure 2. This often results in optimization with inconsistent ground truths, leading to feature-averaged reconstructions. Secondly, it is well established that single-step denoising outputs from diffusion models are often suboptimal, typically lacking in detail and high-frequency components. This limitation further diminishes the performance of RED-diff, resulting in less precise outcomes.

## 3.2 Self-consistency for Inverse Problem Solving

Given the uncertainty and imprecision of DR, the deterministic process can be employed to achieve more consistent and accurate regularization. A straightforward approach is applying reverse ODE inversion (Su et al., 2022) to encode $\mu$ into noisy data, eliminating the need for random perturbations. Alternatively, we can employ a multi-step solver to obtain a more accurate estimate of $\hat{\mu}_{0|t}$ (Tang et al., 2023). While these methods improve estimation accuracy, they come with a significant trade-off in terms of computational cost. Each iteration of gradient descent requires resource-intensive inversions and multi-step solver executions, substantially increasing the overall computation time. The added computational burden can be prohibitive, particularly in applications where efficiency is crucial for solving inverse problems.

We observe that the above process can draw inspiration from recent advances in consistency distillation (CD) (Song et al., 2023b), which enable efficient approximation of diffusion ODE without the need for intensive simulations. CD enhances self-consistency by minimizing the difference of outputs between two adjacent points (e.g., $t$ and $t+\varepsilon$) on the ODE trajectory (Eq. 4). Given that the output at $t$ is more accurate than at $t+\varepsilon$, the model can iteratively propagate the trajectory endpoint back to $t = 1$. Motivated by this concept, we present Consistency Regularization (CR) as the difference between two adjacent outputs. The gradient of this regularization term is expressed as follows:

$$\nabla_\mu \mathcal{L}_{\text{CR}} = \nabla_\mu \|\mathbf{y} - \mathcal{A}(\mu)\|_2^2 + \mathbb{E}_{t,\epsilon}[\hat{\mu}_{0|t+\varepsilon} - \hat{\mu}_{0|t}], \tag{11}$$

where $\hat{\mu}_{0|t+\varepsilon} = \mathbb{E}[\mu|\mu_{t+\varepsilon}]$. The points $\mu_{t+\varepsilon}$ and $\mu_t$ are adjacent, and $\mu_t$ can be computed using the Euler solver as $\mu_t = \alpha_t \hat{\mu}_{0|t+\varepsilon} + \sigma_t \epsilon_\theta(\mu_{t+\varepsilon}, t+\varepsilon)$. The key idea behind Eq. 11 is as follows: the clean image predicted at $t$ has a higher fidelity compared to that at $t+\varepsilon$. By progressively incorporating more complex features through the difference between these two estimates, the optimized variable incrementally converges toward the real image at $t = 0$. We can further simplify the regularization term to operate within the noise domain.

$$\hat{\mu}_{0|t+\varepsilon} - \hat{\mu}_{0|t} = \hat{\mu}_{0|t+\varepsilon} - (\mu_t - \sigma_t \epsilon_\theta(\mu_t, t))/\alpha_t \quad \text{(Bring in } \mu_t)$$
$$= \omega_t(\epsilon_\theta(\mu_t, t) - \epsilon_\theta(\mu_{t+\varepsilon}, t+\varepsilon)). \tag{12}$$

Note that the gradient computation outlined above necessitates two network inferences. To reduce computational costs, we can adopt the descending sampling strategy for timesteps as proposed by Mardani et al. (2024), using the noise prediction from the previous step $\epsilon_\theta(\mu_{t+\varepsilon}, t+\varepsilon)$. As shown in Figure 2, we observe that CR indeed produces more consistent $\hat{\mu}_{0|t}$ compared to DR, and produces intricate structural features at an early stage of optimization.

## 3.3 Hybrid Regularization

The CR method provides sharper and more diverse restoration results; however, we have observed that it occasionally produces oversaturation and artifacts, as illustrated in Figure 3. We hypothesize that this issue arises from the lack of randomness in the noise sampling process within CR. Unlike DR, which introduces fresh noise during the optimization process, CR relies on the noise estimation from previous steps. The importance of random noise in enhancing the robustness and quality of generation has been extensively emphasized in the existing literature on

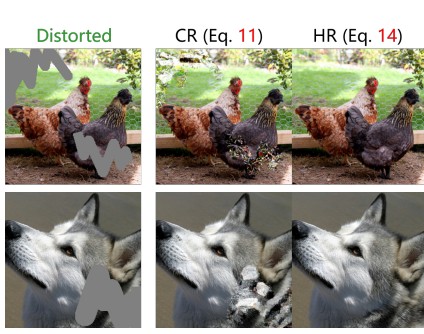

Figure 3: Demonstration of the artifacts and oversaturation present in CR. We effectively mitigate these artifacts by incorporating hybrid noise.

diffusion models (Karras et al., 2022; Xu et al., 2023; Nie et al., 2023). These findings motivate us to explore the possibility of devising a unified framework that remains compatible with these regularization methods, thereby harnessing the strengths of both approaches. In the following, inspired by the study of solvers for diffusion models, we propose a general framework that we dub Hybrid Regularization (HR).

The widely used DDIM solver (Song et al., 2020a) can generate sample $\mathbf{x}_t$ from $\mathbf{x}_{t+\varepsilon}$ using the following expression:

$$\mathbf{x}_t = \alpha_t \underbrace{\hat{\mathbf{x}}_{0:t+\varepsilon}}_{\text{predicted } \mathbf{x}_0} + \underbrace{\sqrt{\sigma_t^2 - \beta_t^2}\boldsymbol{\epsilon}_\theta(\mathbf{x}_{t+\varepsilon}, t+\varepsilon)}_{\text{deterministic noise}} + \underbrace{\beta_t\boldsymbol{\epsilon}}_{\text{random noise}} \tag{13}$$

The DDIM procedure begins with estimating $x_0$ using the network, followed by its immediate projection onto the manifold where the noisy data resides at $t$, according to the forward noising process (Chung et al., 2023b). Note that the noise can be categorized into two parts: deterministic and random noise. Their total variance $(\sqrt{\sigma_t^2 - \beta_t^2})^2 + \beta_t^2 = \sigma_t^2$, where $\beta_t$ governs the stochasticity of the sampling process. Specifically, when $\beta_t = 0$, the sampling process becomes deterministic. Returning to Eqs. 10 and 12, we observe that the disparity between the two gradients stems from the selection of noise: the former utilizes entirely *random noise*, while the latter employs *deterministic noise* predicted by the network. To leverage the benefits of both approaches, we introduce a coefficient $\beta$ to generate a hybrid noise $\boldsymbol{\epsilon}_{\text{hybrid}}$, maintaining constant total variance. Thus, the gradient can be formulated as follows:

$$\nabla_\mu \mathcal{L}_{\text{HR}} = \nabla_\mu \|\mathbf{y} - \mathcal{A}(\mu)\|_2^2 + \mathbb{E}_{t,\epsilon}[\omega_t(\boldsymbol{\epsilon}_\theta(\mu_t, t) - \boldsymbol{\epsilon}_{\text{hybrid}})], \tag{14}$$
$$\boldsymbol{\epsilon}_{\text{hybrid}} \triangleq \sqrt{1-\beta}\boldsymbol{\epsilon}_\theta(\mu_{t+\varepsilon}, t+\varepsilon) + \sqrt{\beta}\boldsymbol{\epsilon}$$

We denote the technique described above as Hybrid Regularization (HR), which constitutes a unified formulation. It is evident that both DR and CR represent extreme cases, where $\beta$ takes values of 1 and 0, respectively. We assume that there exists a *sweet spot* that can effectively balance these two aspects. In the ablation study detailed in Sec. 4.4, we observe that a $\beta$ value of 0.2 is generally sufficient for our method to perform well across a wide range of experiments, obviating the need for task-specific tuning. Unless otherwise specified, we use $\beta = 0.2$ as the default setting.

Intuitively, hybrid noise exhibits a stronger correlation with $\boldsymbol{\epsilon}_\theta(\mu_t, t)$ compared to entirely random noise. Consequently, HR demonstrates reduced gradient variance, thereby promoting more efficient and stable optimization. Meanwhile, incorporating a small degree of stochasticity aids in contract errors accumulated during the inversion process. Another key parameter is the timestep shift $\varepsilon$, for which we found that a value around the $10^{-2}$ order of magnitude works well (Appendix C.8). Our final approach, named **H**ybrid **R**egularization for **D**iffusion-based **I**nverse problem **S**olving (HRDIS), is presented in Algorithm 1, with the Adam optimizer (Kingma & Ba, 2014) employed as the default.

---

**Algorithm 1** Sampling procedure for HRDIS.

---

**Input:** observation $\mathbf{y}$, measurement operator $\mathcal{A}(\cdot)$, number of iterations $N$, timesteps sampling strategy $\{s_n\}_{n=1}^N$, pretrained model $\boldsymbol{\epsilon}_\theta(\cdot, \cdot)$, $\beta$, $\omega_t$
1: Initialize $\mu \leftarrow \mathcal{A}^{-1}(\mathbf{y})$
2: **for** $n = 1, \cdots, N$ **do**
3:      $t \leftarrow s_n$
4:      **if** $n = 1$ **then**
5:          Initialize hybrid noise $\boldsymbol{\epsilon}_{\text{hybrid}} \sim \mathcal{N}(0, \boldsymbol{I})$
6:      **end if**
7:      Forward perturb $\mu_t \leftarrow \alpha_t \mu + \sigma_t \boldsymbol{\epsilon}_{\text{hybrid}}$
8:      Calculate gradient $d_\mu \leftarrow \nabla_\mu \|\mathbf{y} - \mathcal{A}(\mu)\|_2^2 + \omega_t(\boldsymbol{\epsilon}_\theta(\mu_t, t) - \boldsymbol{\epsilon}_{\text{hybrid}})$
9:      Optimize mean $\mu \leftarrow \text{AdamUpdate}(\mu, d_\mu)$
10:      Sample fresh noise $\boldsymbol{\epsilon} \sim \mathcal{N}(0, \boldsymbol{I})$
11:      Calculate hybrid noise $\boldsymbol{\epsilon}_{\text{hybrid}} \leftarrow \sqrt{1-\beta}\boldsymbol{\epsilon}_\theta(\mu_t, t) + \sqrt{\beta}\boldsymbol{\epsilon}$
12: **end for**
**Output:** $\mu$

---

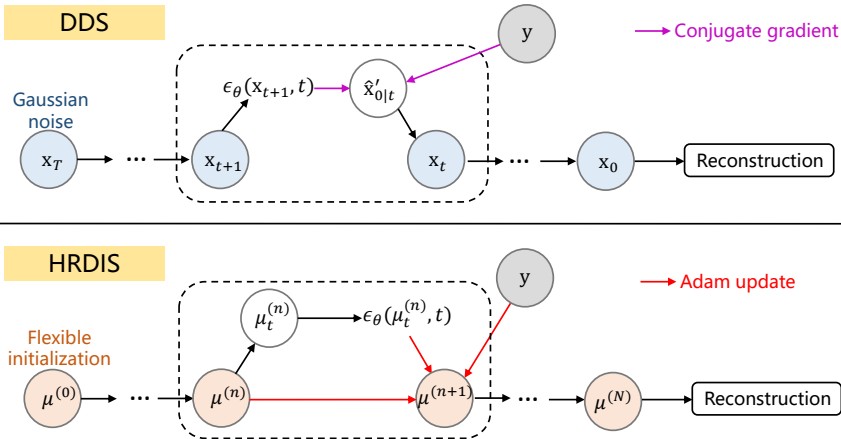

Figure 4: Comparison of DDS (Chung et al., 2023b) (top) and HRDIS (bottom) graph models. HRDIS maintains an additional variable that is optimized during the process, enabling flexible initialization. The superscript $(n)$ denotes the optimization step.

## 3.4 DISCUSSION

In this subsection, we discuss the differences between the proposed HRDIS framework and existing approximate posterior sampling methods, focusing on DDS (Chung et al., 2023b) as a representative example. We compare the graph model of HRDIS with that of DDS in Figure 4. DDS can be understood as approximate posterior sampling, achieved by gradually guiding the unconditional DDIM sampling process. DDS consists of three main steps: 1) Predict $\hat{\mathbf{x}}_{0|t}$ from $\mathbf{x}_t$. 2) Solve the optimization problem: $\min_{\hat{\mathbf{x}}'_{0|t}} \frac{1}{2}\|\mathbf{y} - \mathcal{A}\hat{\mathbf{x}}'_{0|t}\|^2 + \frac{\gamma}{2}\|\hat{\mathbf{x}}'_{0|t} - \hat{\mathbf{x}}_{0|t}\|^2$ using the conjugate gradient (CG). 3) Calculate $\mathbf{x}_{t-1}$ using DDIM. Through these steps, DDS iteratively refines the sampling trajectory $\mathbf{x}_t, \mathbf{x}_{t-1}, \ldots, \mathbf{x}_0$ by solving the subproblem in Step 2. In contrast, HRDIS builds on the RED-diff framework and formulates sampling as a stochastic optimization. A key difference is that HRDIS introduces and maintains an additional variable $\mu$, initialized as $\mathcal{A}^{-1}(\mathbf{y})$, throughout the optimization. This variable is updated using gradients $\nabla_\mu \mathcal{L}_{\mathrm{HR}}$ with an off-the-shelf Adam optimizer.

These fundamental differences result in notably distinct sampling behaviors. DDS gradually transitions from noise to reconstruction. In contrast, HRDIS exhibits an evolution that bridges the degraded image and the reconstruction, reflecting its unique generative dynamics (see Figure 17).

## 4 EXPERIMENTS

We structure our experiments to address the following inquiries. **Q1:** Does our proposed HRDIS effectively mitigate the issue of generating blurry images, as observed in RED-diff (Mardani et al., 2024), while simultaneously producing sharp and diverse reconstruction outcomes? **Q2:** How does the performance of HRDIS compare to that of state-of-the-art diffusion model-based inverse problem solvers, such as $\Pi$GDM (Song et al., 2022), FPS (Dou & Song, 2024), among others? Furthermore, what is the computational efficiency of our approach compared to these alternatives? **Q3:** What is the optimal choice for the hyperparameter $\beta$ in regulating stochasticity within the framework of hybrid regularization?

To address **Q1**, we examine the inversion results of challenging images within the dataset, assessing whether HRDIS can effectively restore high-frequency details compared to RED-diff. For **Q2**, we conduct a quantitative comparison of quality and efficiency against state-of-the-art algorithms. Lastly, to explore **Q3**, we conduct experiments to analyze the hyperparameter $\beta$ selection.

**Experimental Setup:** We assess the effectiveness of the proposed HRDIS method across a range of image restoration tasks, encompassing both linear and nonlinear inverse problems such as image inpainting, super-resolution, compressed sensing (CS), phase retrieval, high dynamic range (HDR) tasks, and nonlinear deblurring. Our experiments are conducted on the ImageNet $256 \times 256$ (Deng et al., 2009) and FFHQ $256 \times 256$ (Karras et al., 2019) datasets, with results derived from 1k validation

Table 1: Quantitative evaluation (FID, LPIPS, CA (unit:%)) of solving *linear* inverse problems on ImageNet 256×256-1k validation dataset. **Bold**: best, underline: second best.

| Method | Inpaint (10-20%) | | | Inpaint (20-30%) | | | SR (×4) | | | CS (25%) | | |
|---|---|---|---|---|---|---|---|---|---|---|---|---|
| | LPIPS ↓ | FID ↓ | CA ↑ | LPIPS ↓ | FID ↓ | CA ↑ | LPIPS ↓ | FID ↓ | CA ↑ | LPIPS ↓ | FID ↓ | CA ↑ |
| DDRM (Kawar et al., 2022) | 0.071 | 13.88 | 73.3 | 0.123 | 26.09 | 70.2 | 0.265 | 47.43 | 65.3 | 0.250 | 59.40 | 51.7 |
| DPS (Chung et al., 2022) | 0.146 | 27.65 | 69.0 | 0.182 | 36.64 | 64.6 | 0.197 | 37.35 | 67.7 | 0.168 | 34.20 | 68.2 |
| ΠGDM (Song et al., 2022) | 0.073 | 12.85 | 73.7 | 0.118 | 21.84 | **72.8** | 0.150 | **28.96** | **71.8** | 0.075 | 22.98 | **73.5** |
| DDNM (Wang et al., 2022) | 0.075 | 14.06 | 72.0 | 0.106 | 26.61 | 67.9 | 0.251 | 47.15 | 61.9 | 0.247 | 53.16 | 58.4 |
| GDP (Fei et al., 2023) | 0.070 | 13.91 | 72.5 | 0.135 | 27.83 | 67.3 | 0.234 | 44.08 | 62.4 | 0.283 | 60.01 | 52.3 |
| DDS (Chung et al., 2023b) | 0.067 | 13.12 | 73.8 | 0.130 | 27.64 | 67.9 | 0.198 | 41.62 | 62.8 | 0.270 | 58.41 | 55.1 |
| FPS (Dou & Song, 2024) | 0.065 | 13.07 | 73.9 | 0.121 | 24.90 | 70.5 | 0.189 | 34.88 | 67.8 | 0.124 | 32.76 | 67.0 |
| RED-diff (Mardani et al., 2024) | 0.067 | 13.20 | 73.6 | 0.117 | 24.67 | 69.5 | 0.249 | 44.16 | 65.8 | 0.108 | 29.90 | 69.3 |
| HRDIS (Ours) | **0.054** | **10.94** | **74.7** | **0.096** | **20.10** | 71.3 | **0.137** | 33.01 | 68.8 | **0.059** | **22.04** | 72.3 |

images, consistent with previous research standards (Chung et al., 2022; Mardani et al., 2024). We utilize pre-trained diffusion models from (Dhariwal & Nichol, 2021) and (Choi et al., 2021). Our comparative analysis includes benchmark techniques, namely DDRM (Kawar et al., 2022), DPS (Chung et al., 2022), ΠGDM (Song et al., 2022), FPS (Dou & Song, 2024) and RED-diff (Mardani et al., 2024). Additional details on the experimental setup can be found in the Appendix B.2.

## 4.1 LINEAR INVERSE PROBLEMS

We begin our experiments with image inpainting, using the *freeform* masks provided by (Saharia et al., 2022a). Specifically, we apply *10%-20%* and *20%-30%* masks on the ImageNet dataset and employ the more challenging *30%-40%* mask on FFHQ. For the super-resolution experiments, we utilize average pooling to perform 4× downsampling on ImageNet and 16× downsampling on FFHQ. In the compressed sensing (CS) task, we adopt an orthogonal sampling matrix applied to the image blocks, with a sampling rate of 25% for ImageNet and 10% for FFHQ. We evaluate our results using two widely adopted metrics: Learned Perceptual Im-

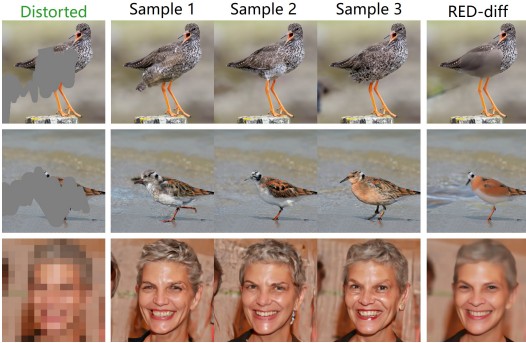

Figure 5: Diversity of reconstructions generated by HRDIS (Columns 2-4).

age Patch Similarity (LPIPS) (Zhang et al., 2018) and Fréchet Inception Distance (FID) (Heusel et al., 2017), computed on the test images. Additionally, for the ImageNet dataset, we report classifier accuracy (CA) using a pre-trained ResNet50 (He et al., 2016).

Tables 1 and 2 present the quantitative outcomes of linear inverse problem solving on ImageNet and FFHQ, respectively. It is evident that HRDIS consistently ranks within the top two across all metrics and significantly outperforms other methods in numerous instances. Particularly noteworthy is HRDIS's exceptional performance in the inpainting and CS task, where it outshines all other techniques. While ΠGDM demonstrates superiority in the super-resolution task, HRDIS closely follows as the runner-up. Following the quantitative analysis, we present a visual comparison with RED-diff in Figure 5. It is evident that RED-diff frequently converges to the blurry images, which may not always align with the desired inversion result. In contrast, HRDIS exhibits the ability to generate diverse and plausible restored images across various random seed settings, offering a more flexible and robust solution to the image restoration task. Figures 6 and 7 provides a qualitative comparison with other state-of-the-art inversion methods. Our observations indicate that DDRM tends to generate less realistic results and struggles to guide the reverse diffusion process toward achieving globally harmonious outcomes. Additionally, ΠGDM occasionally exhibits instability, particularly evident in challenging samples, attributed to the necessity of backpropagating through the score network, resulting in failures in certain instances. In contrast, HRDIS can effectively recognize context and produce better restorations. See Appendix C.9 for more visualizations.

The wall-clock time and GPU memory of the different algorithms for the inpainting are reported in Table 3. Our observations reveal that the optimization step of our method can be an order of magnitude fewer than that of RED-diff and FPS. In comparison to state-of-the-art guidance-based methods (DPS, ΠGDM), our approach demonstrates exceptional lightweightness and memory efficiency. Although

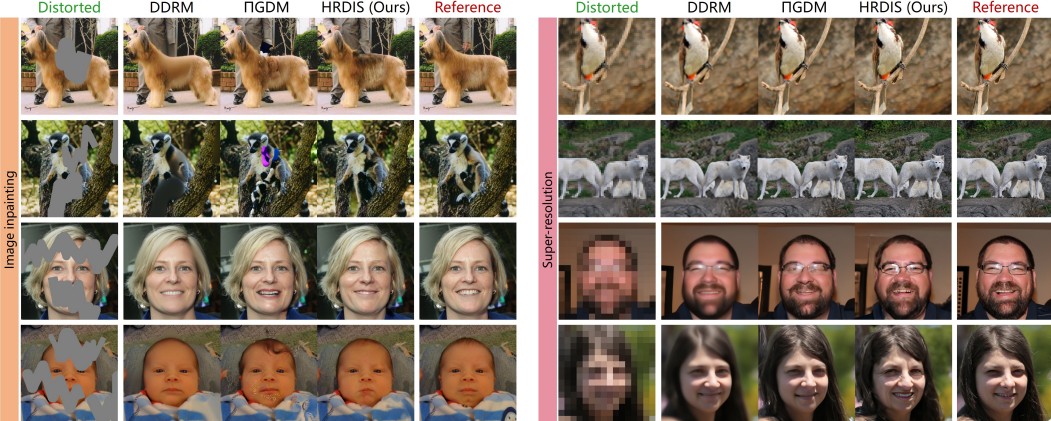

Figure 6: Comparison of the proposed HRDIS with alternatives for inpainting (left) and super-resolution (right) on ImageNet $256 \times 256$ and FFHQ $256 \times 256$.

Table 2: Quantitative evaluation (FID, LPIPS) of solving *linear* inverse problems on FFHQ 256×256-1k validation dataset.

| Method | Inpaint (30-40%) | | SR (×16) | | CS (10%) | |
|---|---|---|---|---|---|---|
| | LPIPS ↓ | FID ↓ | LPIPS ↓ | FID ↓ | LPIPS ↓ | FID ↓ |
| DDRM | 0.099 | 23.91 | 0.324 | 91.44 | 0.535 | 130.2 |
| DPS | 0.138 | 44.75 | 0.220 | 42.31 | 0.228 | 68.50 |
| ΠGDM | 0.084 | 15.84 | **0.188** | **39.33** | 0.091 | 33.25 |
| DDNM | 0.103 | 19.72 | 0.210 | 41.50 | 0.154 | 56.15 |
| GDP | 0.112 | 24.38 | 0.237 | 43.51 | 0.267 | 91.84 |
| DDS | 0.109 | 22.12 | 0.213 | 41.81 | 0.256 | 85.96 |
| FPS | 0.093 | 21.15 | 0.228 | 45.90 | 0.179 | 60.73 |
| RED-diff | 0.086 | 17.38 | 0.286 | 73.39 | 0.184 | 62.19 |
| HRDIS (Ours) | **0.082** | **15.42** | 0.213 | 40.97 | **0.088** | 35.27 |

Table 3: Comparison of wall-clock time and memory consumption, measured on a single RTX 3090 GPU.

| Method | ImageNet | | FFHQ | |
|---|---|---|---|---|
| | Time (s/img) | Memory | Time (s/img) | Memory |
| DDRM | 10 | 4.4G | 4 | 2.5G |
| DPS | 274 | 8.7G | 56 | 6.0G |
| ΠGDM | 33 | 8.8G | 6 | 6.1G |
| DDNM | 13 | 4.6G | 5 | 2.7G |
| GDP | 84 | 4.3G | 33 | 2.4G |
| DDS | 13 | 4.6G | 5 | 2.7G |
| FPS | 95 | 5.2G | 35 | 3.3G |
| RED-diff | 82 | 4.3G | 32 | 2.4G |
| HRDIS (Ours) | 14 | 4.3G | 5 | 2.4G |

DDRM is also efficient, its applicability is limited to linear operator $\mathcal{A}(\cdot)$, and its efficiency is not guaranteed in scenarios where fast singular value decomposition is not feasible.

## 4.2 NONLINEAR INVERSE PROBLEMS

We evaluate the nonlinear inverse problem on the FFHQ dataset, beginning with phase retrieval. Given the inherent instability of phase recovery, we follow the strategy employed by Chung et al. (2022), utilizing an over-sampling rate of 2.0 and reporting the best results under four random seeds. Next, we address the High Dynamic Range (HDR) task, which incorpo-

Table 4: Quantitative evaluation (FID, LPIPS) of solving *nonlinear* inverse problems on FFHQ 256×256-1k validation dataset.

| Method | Phase retrieval | | HDR | | Nonlinear deblurring | |
|---|---|---|---|---|---|---|
| | LPIPS ↓ | FID ↓ | LPIPS ↓ | FID ↓ | LPIPS ↓ | FID ↓ |
| DPS | 0.387 | 54.64 | 0.407 | 84.64 | 0.279 | 52.58 |
| RED-diff | 0.462 | 62.47 | 0.063 | 18.20 | 0.329 | 80.13 |
| HRDIS (Ours) | **0.089** | **38.34** | **0.044** | **13.78** | **0.236** | **52.36** |

rates a truncation function to crop pixel values, represented as $\mathcal{A}(\mathbf{x}) = \text{Clip}(2\mathbf{x}, -1, 1)$. we adopt a pre-trained network to simulate the blurring degradation operator as described in Tran et al. (2021). Since methods such as DDRM, ΠGDM, and FPS are not scalable to these nonlinear challenges, we compare our approach exclusively with DPS and RED-diff.

Table 4 presents the quantitative metrics for various solvers applied to the nonlinear inverse problem. Notably, the LPIPS scores for phase retrieval are significantly higher than those of the baseline, demonstrating a substantial improvement. Figures 7 and 8 provide qualitative results. We can see that in the phase retrieval task, HRDIS generates reconstructions that closely match the reference image, whereas DPS and RED-diff yield unrealistic outputs. Additionally, DPS encountered challenges with the HDR task, failing to reconstruct the original image accurately. For the nonlinear deblurring task, the RED-diff method exhibited significant gradient variance, leading to a pronounced loss of detail and reduced fidelity. In contrast, our observations indicate that HRDIS consistently generates highly realistic samples, even in these more challenging nonlinear scenarios.

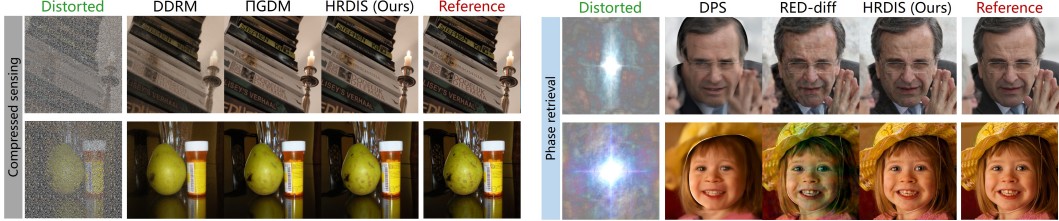

Figure 7: Comparison of the proposed HRDIS with alternatives for compressed sensing (left) and phase retrieval (right) on ImageNet $256 \times 256$ and FFHQ $256 \times 256$.

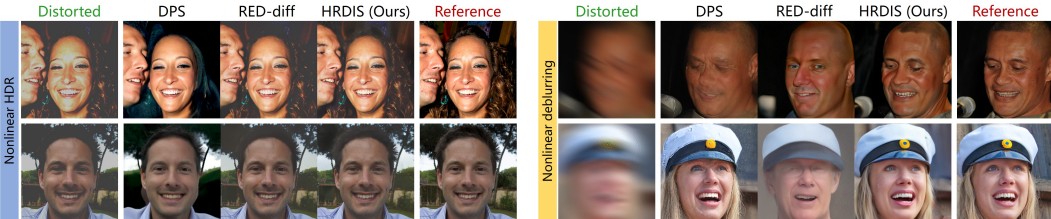

Figure 8: Comparison of the proposed HRDIS with alternatives for HDR task (left) and nonlinear deblurring (right) on FFHQ $256 \times 256$.

### 4.3 HANDLING VARIOUS NOISE STATISTICS

As an optimization-based framework, HRDIS is adept at managing various types of noise. In this subsection, we empirically evaluate HRDIS's performance under noisy conditions. To assess its robustness, we apply a freeform mask for inpainting and introduce three types of noise: Gaussian, Poisson, and speckle, into the observations. The results, detailed in Appendix C.2, show that HRDIS consistently outperforms other methods in most scenarios. This demonstrates the framework's robustness and adaptability in handling various measurement statistics.

### 4.4 ABLATIONS: INFLUENCE OF $\beta$

We conduct an ablation analysis to investigate the impact of the parameter $\beta$ within the HRDIS method, focusing on inpainting and super-resolution using the ImageNet dataset, as well as nonlinear deblurring on the FFHQ dataset. The quantitative results in Figure 9 indicate that a $\beta$ value of 0.2 consistently achieves lower LPIPS and FID scores across different tasks. As shown in Figure 11 (Appendix C.1), excessively high

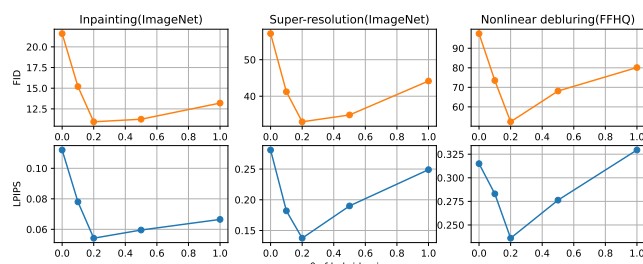

Figure 9: Quantitative results for varying values of $\beta \in \{0, 0.1, 0.2, 0.5, 1\}$.

values of $\beta$ result in over-smoothing, while very low values introduce noticeable artifacts. Notably, $\beta = 0.2$ strikes an optimal balance, effectively reconstructing plausible high-frequency details.

### 5 CONCLUSION

This paper addresses the challenges associated with solving diffusion-based inverse problems using Denoising Regularization (DR). To tackle these challenges, we introduce the Consistency Regularization (CR) method, which effectively mitigates the issue of inaccurate gradient estimations. Additionally, we explore the integration of hybrid noise, resulting in the development of a unified HRDIS framework that fosters synergy between the two regularization techniques. The proposed framework is versatile, making it applicable to both linear and nonlinear inverse problems, as well as accommodating diverse measurement statistics. Comprehensive experimental evaluations demonstrate that HRDIS not only surpasses current state-of-the-art methods but also maintains high computational efficiency.

## ACKNOWLEDGEMENTS

This work was supported by the National Natural Science Foundation of China under Grant 62325101, 62031001, and also supported by the National Key Laboratory of Unmanned Aerial Vehicle Technology in NPU under Grant WR202403, and supported by the Fundamental Research Funds for the Central Universities.

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

# A DISCUSSION

## A.1 IMPACT STATEMENT

Our proposed framework presents a novel approach to addressing inverse problems by applying diffusion models. Given its versatility and computational efficiency, we anticipate that our framework will benefit the relevant research community. However, it is crucial to acknowledge the potential misuse of our approach in generating deceptive or malicious content. Additionally, reliance on generative techniques like diffusion models raises concerns regarding inherent biases present in the vast datasets utilized for training. These biases have the potential to manifest in the generated outputs. It may be necessary to establish ethical guidelines and regulations to ensure responsible development and deployment of relevant technologies.

## A.2 LIMITATIONS AND FUTURE WORKS.

While our proposed framework demonstrates efficacy in addressing both linear and nonlinear scenarios, its primary limitation arises from the requirement to ascertain the form of the measurement operator or the computational procedure. Consequently, an enticing avenue for future exploration is the extension of our approach to tackling blind inverse problems (Chung et al., 2023a; Murata et al., 2023), where such prior knowledge is unavailable. We provide a preliminary experiment in Appendix C.7. Furthermore, considering the prevalence of diffusion models operating within latent spaces (Rombach et al., 2022), a promising direction for advancement involves extending our methodology to encompass latent diffusion models. Finally, since HRDIS involves stochastic optimization for non-convex problems, its convergence behavior remains an open theoretical question and warrants further investigation.

## A.3 RELATED WORKS

Recent research has delved into the utilization of diffusion generation for solving inverse problems preemptively. Earlier studies (Song & Ermon, 2019; Song et al., 2020b; Lugmayr et al., 2022) focused on substituting observational components within the generation process to facilitate tasks like image inpainting. (Song et al., 2021) applied this technique to medical image reconstruction, while DDRM (Kawar et al., 2022) extended it to address more general degeneracies through singular value decomposition. Meanwhile, ILVR (Choi et al., 2021) achieved reference-based generation using a low-pass filter. DDNM (Wang et al., 2022) tackled inverse problem-solving by refining only the zero space during generation. Despite these advancements, these methods often remain confined to linear degeneracies and struggle with producing highly realistic generation results.

Subsequent research, such as DPS (Chung et al., 2022) and ΠGDM (Song et al., 2022), has aligned the generation process with observations by incorporating a guidance term of likelihood, resulting in clearer predictions. However, a drawback of these approaches is the computational expense and instability associated with backpropagating through the score network. Additionally, Zhu et al. (2023) combined diffusion modeling with Half-Quadratic-Splitting for plug-and-play image restoration. There has also been a series of works (Trippe et al., 2022; Wu et al., 2024b; Dou & Song, 2024; Cardoso et al., 2023) has also linked a posteriori sampling of diffusion models to sequential Monte Carlo. These methods typically maintain multiple particles during the generation process and therefore require more memory. PSLD (Rout et al., 2024) and ReSample Song et al. (2023a) explored the use of latent diffusion to solve the inverse problem. More recently, RED-diff (Mardani et al., 2024) offers a new perspective on diffusion-based inverse problem solving by modeling it as a variational problem. Despite its efficacy, RED-diff suffers from mode-collapse. In response to this limitation, we introduce the HRDIS framework in this paper, building upon RED-diff to significantly enhance the accuracy and efficiency of image inversion. Xu et al. (2024) discusses the use of consistency models to improve DPS, although this approach may be limited by the fact that consistency model checkpoints are not widely available.

Some text-to-3D techniques (Poole et al., 2022; Shi et al., 2023; Liang et al., 2024; Wang et al., 2024) are also pertinent to our work, as they commonly optimize the score distillation loss to generate 3D assets from a 2D prior. This loss bears resemblance to RED-diff. However, our approach diverges in its objective. While text-to-3D efforts focus on lifting a 2D prior to 3D, our focus lies in leveraging a

diffusion prior to solving inverse problems. Thus, our methodology differs fundamentally in purpose, despite similarities in the optimization paradigm.

## B  ADDITIONAL METHOD DETAILS

### B.1  ADDITIONAL JUSTIFICATION FOR CR

We aim to leverage deterministic probability flow to enhance Eq. 10, where the objective is closely linked to the distillation of diffusion models, as both tasks involve fitting the endpoint of the PF ODE provided by the pre-trained diffusion model. However, a nuanced distinction exists between them: Eq. 10 optimizes the *image* $\mu$, whereas the distillation process involves optimizing the *parameters* $\varphi$ of the distillation network $f_\varphi(\cdot, \cdot)$. Specifically, the typical procedure for distilling a pre-trained diffusion model involves simulating an inverse PF ODE to collect $\mathbf{x}_0 \sim p_{\text{data}}$, which yields training data for distillation. Subsequently, the following loss function is employed for distillation:

$$\min_\varphi \|f_\varphi(\mathbf{x}_t, t) - \mathbf{x}_0\|_2^2. \tag{15}$$

To circumvent the need for extensive simulation, Song et al. (2023b) introduces an innovative alternative approach termed Consistency Distillation (CD). Their study reveals the *self-consistency property* within the ODE trajectories of the diffusion model, wherein points along the same trajectory correspond to identical initial points at $t = 0$ (Figure 10 (a)). Therefore, they propose an indirect distillation method by minimizing predictions at adjacent times (e.g., $t$ and $t + \varepsilon$) for single-step generation,

$$\min_\varphi \|f_\varphi(\mathbf{x}_{t+\varepsilon}, t + \varepsilon) - \text{sg}(f_\varphi(\mathbf{x}_t, t))\|_2^2, \tag{16}$$

where $f_\varphi(\mathbf{x}_{t+\varepsilon}, t + \varepsilon)$ and $f_\varphi(\mathbf{x}_t, t)$ can be viewed as student and teacher predictions, respectively. Since the boundary condition $f_\varphi(\mathbf{x}_0, 0) = \mathbf{x}_0$ is satisfied, the prediction $f_\varphi(\mathbf{x}_t, t)$ is closer to the ground truth than that at $t + \varepsilon$. Through iterative propagation, the model aligns the endpoint of the trajectory such that, for any $t$, $f_\varphi(\mathbf{x}_t, t) \approx \mathbf{x}_0$.

In the context of diffusion-based inverse problem solving, we aim to penalize $\|\mu - \mathbf{x}_0\|_2^2$ for regularization (Figure 10 (b)). However, obtaining the ideal $x_0$ is often challenging, as it may require inversion and solving the ODE (Su et al., 2022; Liang et al., 2024). Notably, adjacent points of $\mu_t$ are easier to obtain. Thus, we incorporate insights from CD, which rely on the difference between outputs at two points along the same trajectory to iteratively refine $\mu$. This process enables us to indirectly align $\mu$ with the endpoint of the ODE.

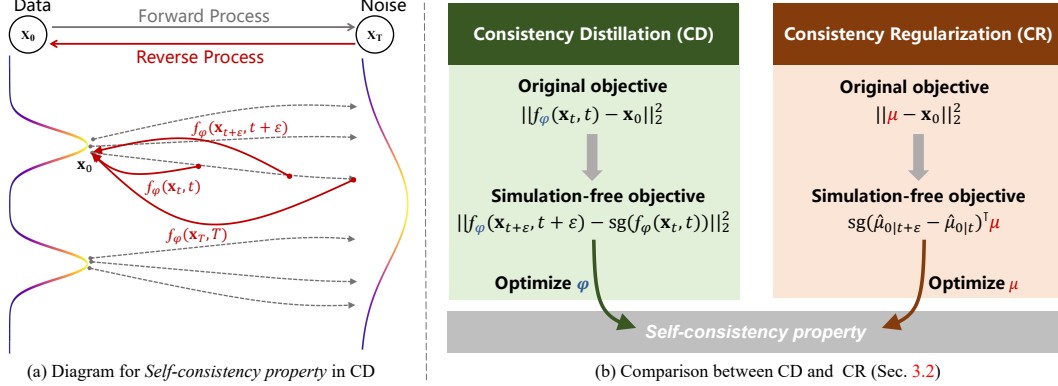

(a) Diagram for *Self-consistency property* in CD      (b) Comparison between CD and  CR (Sec. 3.2)

Figure 10: Schematic illustrating the motivation for Consistency Regularization (CR). **(a)** Consistency Distillation (CD) (Song et al., 2023b) is trained to map points on any ODE trajectory (gray dashed line) of diffusion models to the trajectory's origin in a single step, ensuring the *self-consistency property* is maintained. **(b)** CR and CD share the same concept of transforming the original objective into a simulation-free objective based on *self-consistency*. However, they differ in that the former optimizes $\mu$ while the latter optimizes parameter $\varphi$.

Table 5: Hyperparameter choice for the proposed method.

| Problem | Inpainting | SR | CS | Phase retrieval | HDR | Nonlinear deblurring |
|---|---|---|---|---|---|---|
| $N$ | 150 | 100 | 300 | 500 | 100 | 100 |
| $\lambda_1$ | 1 | 1 | 1 | 0.5 | 1 | 0.02 |
| $\lambda_2$ | 0.25 | 0.20 | 0.20 | 0.25 | 0.25 | 0.25 |
| learning rate | 0.1 | 0.5 | 0.5 | 0.7 | 0.1 | 0.1 |

## B.2 EXPERIMENTAL SETUP

All experiments, including time and memory calculations, were conducted on a single NVIDIA RTX 3090 GPU. We employed the Adam optimizer (Kingma & Ba, 2014) with momentum parameters set to (0.9, 0.99). The parameter $\beta$, utilized in synthesizing the hybrid noise, remained fixed at 0.2 throughout our experiments. We also chose descending timestep from $t = 1$ to $t = 0$ as in (Mardani et al., 2024). The denoiser weight $\omega_t$ is determined as the inverse signal-to-noise ratio (SNR), $\sigma_t/\alpha_t$.

In practice, we introduce two coefficients to balance the reconstruction and regularization terms,

$$\nabla_\mu \mathcal{L}_{\text{HR}} = \lambda_1 \nabla_\mu \|\mathbf{y} - \mathcal{A}(\mu)\|_2^2 + \lambda_2 \mathbb{E}_{t,\varepsilon,\epsilon}[\omega_t(\epsilon_\theta(\mu_t, t) - \epsilon_{\text{hybrid}})], \tag{17}$$

For most tasks, optimizing the number of steps in the range of $N = 100 \sim 150$ produces satisfactory results. For more challenging degenerations, such as compressed sensing and phase retrieval, we use more steps to improve performance further. In addition, for the phase retrieval task, which is particularly sensitive to initial noise, we found that starting with random noise for the first 250 steps, followed by hybrid noise for the remaining 250 steps, significantly improves performance. Table 5 details the selected hyperparameters of our proposed method.

Below, we give the Pytorch-style pseudocode for our HRDIS implementation.

```python
# Add noise to perturbation 'mu'
noise_xt = torch.randn_like(mu)
if step == 0:
    hybrid_noise = noise_xt
else:
    # Synthesize hybrid noise
    hybrid_noise = (1 - beta).sqrt() * deter_noise + beta.sqrt() *
    noise_xt
xt = alpha_t.sqrt() * mu + (1 - alpha_t).sqrt() * hybrid_noise

# Call the denoising model to get the noise 'et'
with torch.no_grad():
    et = model(xt, t).detach()

# Compute reconstruction and regularization terms
e_obs = y_0 - A(mu)
loss_obs = (e_obs**2).mean() / 2
loss_noise = torch.mul((et - hybrid_noise).detach(), mu).mean()

# Compute the weights of two items
snr_inv = (1 - alpha_t).sqrt() / alpha_t.sqrt()
v_t = lambda_1
w_t = lambda_2 * snr_inv
loss = w_t * loss_noise + v_t * loss_obs

# Adam step for 'mu'
optimizer.zero_grad()
loss.backward()
optimizer.step()

# Store the noise 'et' for the next iteration
deter_noise = et.clone()
step += 1
```

Listing 1: Pseudocode of HRDIS for performing one optimization step.

For the implementation of baseline methods, including DDRM (Kawar et al., 2022), DPS (Chung et al., 2022), DDNM (Wang et al., 2022), GDP (Fei et al., 2023), DDS (Chung et al., 2023b), FPS (Dou & Song, 2024), and RED-diff (Mardani et al., 2024), we utilized the official repositories provided by the respective authors. However, since no official implementation of ΠGDM (Song et al., 2022) was available, we faithfully reproduced it using the pseudo-code provided by the authors. For the hyperparameters, we primarily adhered to the original configurations, with slight fine-tuning to achieve optimal performance. DDRM and ΠGDM were configured with 100 steps, while DPS, FPS, and RED-diff required 1000 steps for effective performance. Additionally, we observed that increasing the number of particles in FPS yielded only marginal improvements, so we set it to 1 for our experiments.

We use the default code and settings of each competitor from their official homepages as below.

- DDRM (Kawar et al., 2022): https://github.com/bahjat-kawar/ddrm
- DPS (Chung et al., 2022): https://github.com/DPS2022/diffusion-posterior-sampling
- DDNM (Wang et al., 2022): https://github.com/wyhuai/DDNM
- GDP (Fei et al., 2023): https://github.com/Fayeben/GenerativeDiffusionPrior
- DDS (Chung et al., 2023b): https://github.com/HJ-harry/DDS
- FPS (Dou & Song, 2024): https://github.com/ZehaoDou-official/FPS-SMC-2023
- RED-diff (Mardani et al., 2024): https://github.com/NVlabs/RED-diff

## C  ADDITIONAL RESULTS

### C.1  IMPACT OF $\beta$ FOR HYBRID NOISE

We present the effect of varying $\beta$ on the outputs of different inverse problems in Figure 11. When $\beta$ is set to 0, corresponding to the CR described in Sec. 3.2, we observe the generation of high-frequency information, albeit accompanied by severe artifacts. When $\beta$ is set to 1, the result aligns with the DR used in RED-diff (Mardani et al., 2024), producing a blurry solution. Notably, when $\beta$ is set to 0.2, we identify a sweet spot that effectively balances the two aforementioned aspects, thus unlocking their potential for synergy.

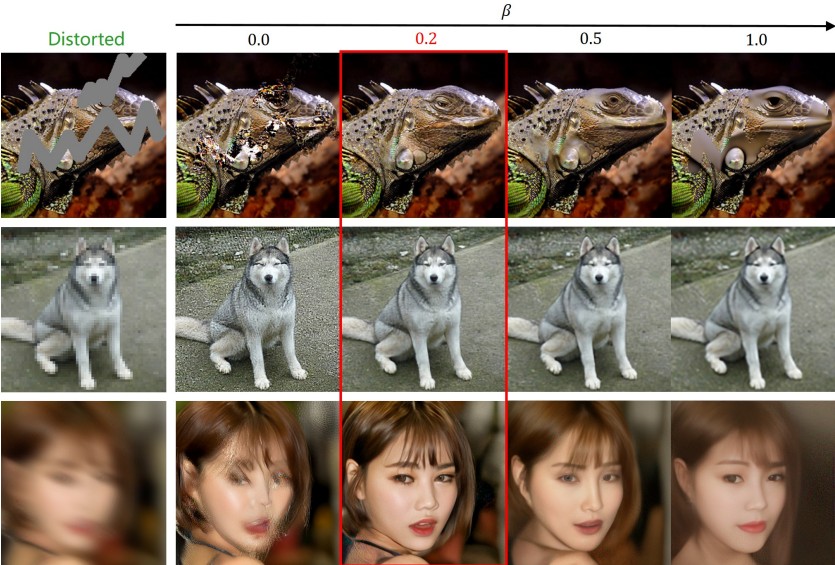

Figure 11: Restoration results for various inverse problems under different $\beta$.

## C.2 NOISY INVERSE PROBLEMS

In this section, we empirically verify the performance of HRDIS under noisy observation conditions. Specifically, we simulate four types of noise: Gaussian noise with standard deviations of 0.05 and 0.1, Poisson noise with the noise level set to 1.0, and speckle noise with a standard deviation of 0.1.

As an optimization-based framework, HRDIS accommodates noise without requiring modifications to the algorithm. For Gaussian noise with a standard deviation of 0.05, we set the weight $\lambda_1$ of the reconstruction term to 0.1, and for other noise types, we reduce $\lambda_1$ to 0.05. The ablation study in RED-diff (Mardani et al., 2024) indicated that the method performs optimally when the time-dependent parameter $\omega_t$ is set to $\frac{1}{\text{SNR}_t} := \sigma_t/\alpha_t$. While this configuration is effective for HRDIS under noiseless conditions, it presents challenges in noisy settings, as the decreasing $\omega_t$ tends toward 0, leading to insufficient regularization and potential overfitting to noisy data. To address this issue, we implemented a clipping mechanism for $\omega_t$ (i.e., torch.clip $\left(\frac{1}{\text{SNR}_t}, \min = 2.0\right)$), ensuring that the regularization term remains effective throughout the optimization process.

The quantitative results presented in Tables 6 and 7 demonstrate that HRDIS consistently outperforms other methods across most scenarios. Furthermore, qualitative comparisons in Figures 7 to 15 reveal that $\Pi$GDM is particularly susceptible to noise, whereas DDRM, though effective at noise removal, often results in blurred reconstructions. Additionally, DPS exhibits instability and is prone to producing artifacts in the presence of Poisson and speckle noise. Overall, our method exhibits strong robustness to various types of noise, confirming its effectiveness across different measurement statistics.

Table 6: Quantitative evaluation (FID, LPIPS, CA (unit:%)) of solving inpainting problems under Gaussian noise on ImageNet 256×256 and FFHQ 256×256 validation dataset. **Bold**: best, underline: second best.

|  | Method | ImageNet 256×256 | | | FFHQ 256×256 | |
|---|---|---|---|---|---|---|
|  |  | LPIPS ↓ | FID ↓ | CA ↑ | LPIPS ↓ | FID ↓ |
| $\gamma = 0.05$ | DDRM (Kawar et al., 2022) | **0.109** | 30.1 | 70.1 | **0.138** | 45.1 |
|  | $\Pi$GDM (Song et al., 2022) | 0.203 | 35.9 | 66.3 | 0.236 | 44.7 |
|  | RED-diff (Mardani et al., 2024) | 0.143 | 30.8 | 66.8 | 0.187 | 55.9 |
|  | HRDIS (Ours) | 0.129 | **29.4** | **70.4** | 0.153 | **31.1** |
| $\gamma = 0.1$ | DDRM (Kawar et al., 2022) | **0.173** | 50.2 | 63.4 | 0.179 | 59.3 |
|  | $\Pi$GDM (Song et al., 2022) | 0.412 | 62.8 | 52.0 | 0.440 | 69.6 |
|  | RED-diff (Mardani et al., 2024) | 0.278 | 47.9 | 60.3 | 0.340 | 91.5 |
|  | HRDIS (Ours) | 0.192 | **38.3** | **65.4** | **0.159** | **31.5** |

Table 7: Quantitative evaluation under Poisson and speckle noise on FFHQ 256×256-1k validation dataset. **Bold**: best, underline: second best.

| Method | Poisson | | Speckle | |
|---|---|---|---|---|
|  | LPIPS ↓ | FID ↓ | LPIPS ↓ | FID ↓ |
| DPS Chung et al. (2022) | 0.226 | 73.09 | 0.240 | 85.41 |
| RED-diff Mardani et al. (2024) | 0.192 | 58.96 | 0.232 | 78.72 |
| HRDIS (Ours) | **0.148** | **29.13** | **0.152** | **30.75** |

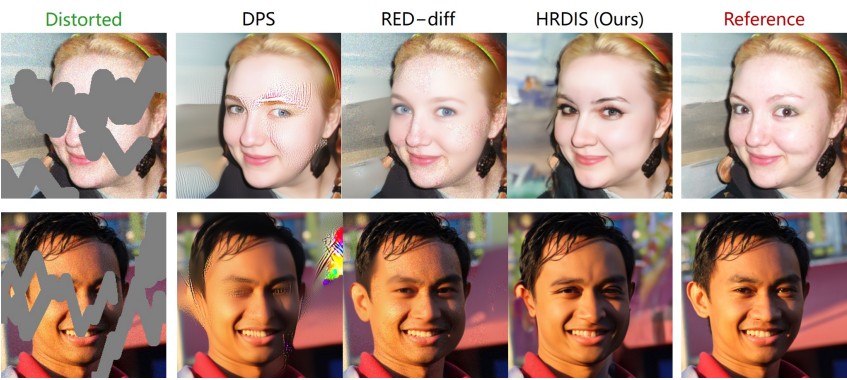

Figure 12: Comparison of the proposed HRDIS with alternatives under Poisson noise, including DPS (Chung et al., 2022), RED-diff (Mardani et al., 2024) and HRDIS (Ours).

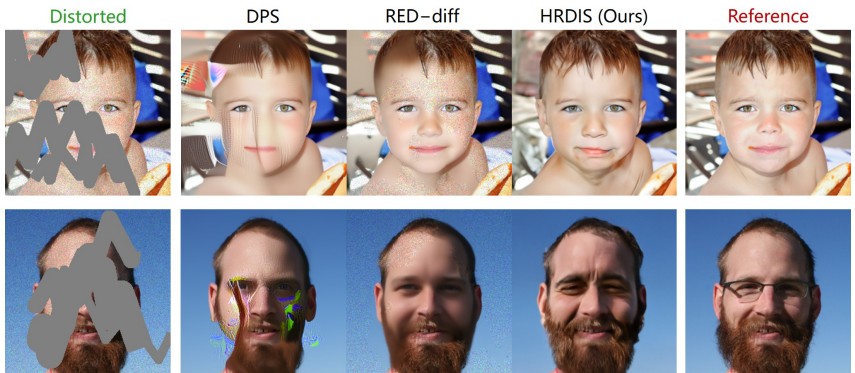

Figure 13: Comparison of the proposed HRDIS with alternatives under speckle noise, including DPS (Chung et al., 2022), RED-diff (Mardani et al., 2024) and HRDIS (Ours).

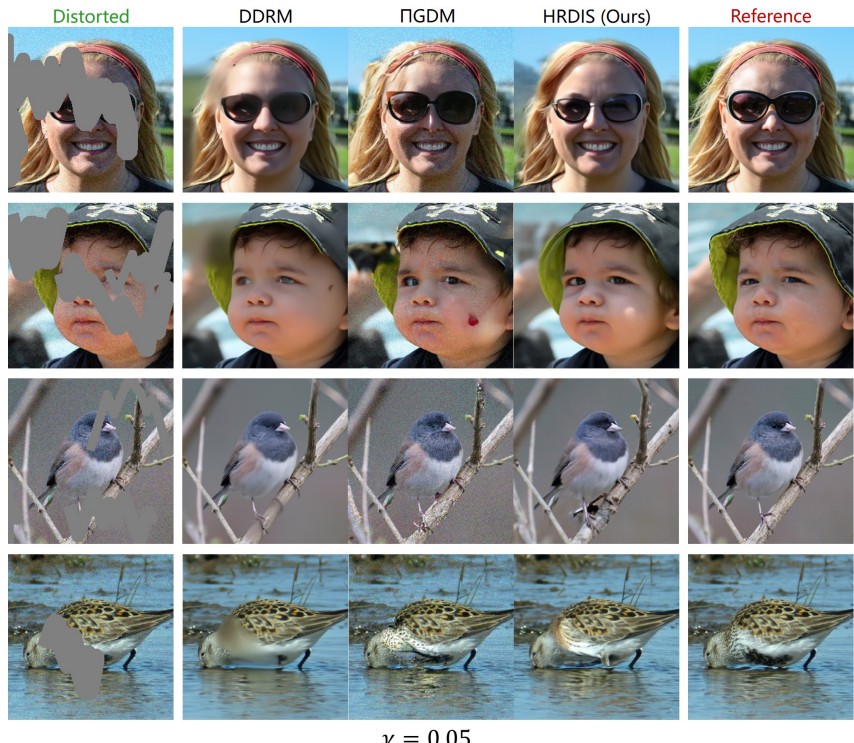

Figure 14: Comparing methods for inpainting problem with Gaussian noise ($\gamma = 0.05$), including DDRM (Kawar et al., 2022), ΠGDM (Song et al., 2022) and HRDIS (Ours).

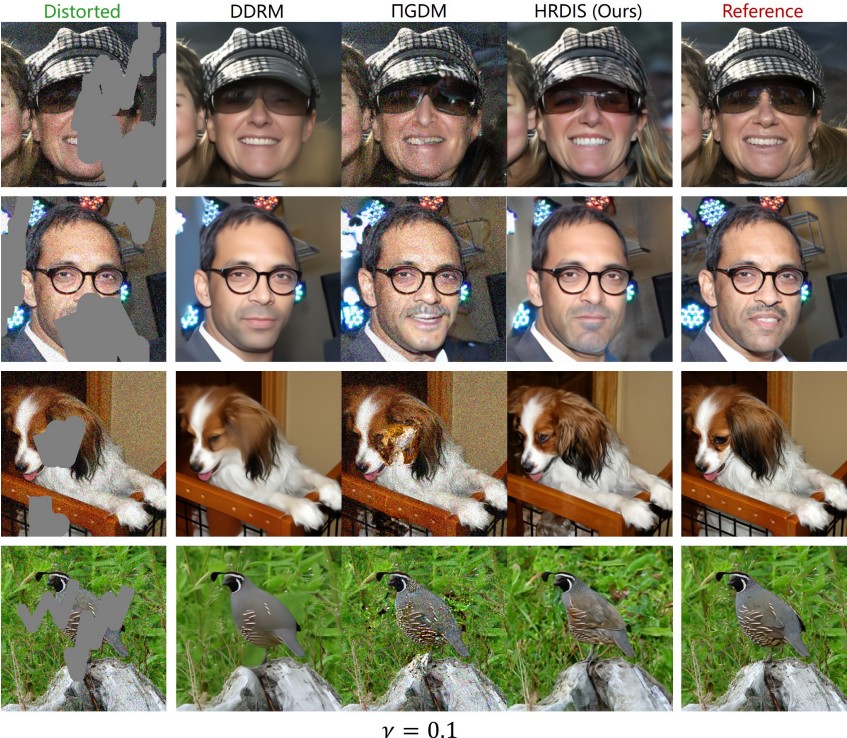

Figure 15: Comparing methods for inpainting problem with Gaussian noise ($\gamma = 0.1$), including DDRM (Kawar et al., 2022), ΠGDM (Song et al., 2022) and HRDIS (Ours).

### C.3 ABLATION FOR COMBINATION OF DR AND DDIM SOLVER

A straightforward approach is to incorporate existing DDIM solvers (Song et al., 2020a) using hybrid noise to enhance DR. In this subsection, we evaluate this approach. The quantitative results, presented in Table 8, indicate that incorporating DDIM into DR improves the performance of RED-diff, aligning with the conclusions from Sec. 3.1. While the random perturbations in RED-diff introduce uncertainty, the DDIM solver mitigates this issue to some extent. However, DR+DDIM still falls short compared to HRDIS. By providing a smooth interpolation between DR and CR, HRDIS forms a more flexible and synergistic framework. Figure 16 illustrates qualitative comparisons that further highlight the advantages of HRDIS.

Table 8: Ablation for the combination of DR and DDIM across these tasks: ImageNet256×256-Inpainting, ImageNet256×256-Super Resolution, and FFHQ256×256-Nonlinear Deblurring.

| Method | Inpainting | | | Super Resolution | | | Nonlinear Deblurring | |
|---|---|---|---|---|---|---|---|---|
| | LPIPS ↓ | FID ↓ | CA ↑ | LPIPS ↓ | FID ↓ | CA ↑ | LPIPS ↓ | FID ↓ |
| DR | 0.117 | 24.6 | 69.5 | 0.249 | 44.2 | 65.8 | 0.329 | 80.1 |
| DR+DDIM | 0.113 | 23.5 | 70.4 | 0.141 | 35.6 | 65.7 | 0.295 | 73.7 |
| HRDIS (Ours) | **0.096** | **20.1** | **71.3** | **0.138** | **33.0** | **68.8** | **0.236** | **52.4** |

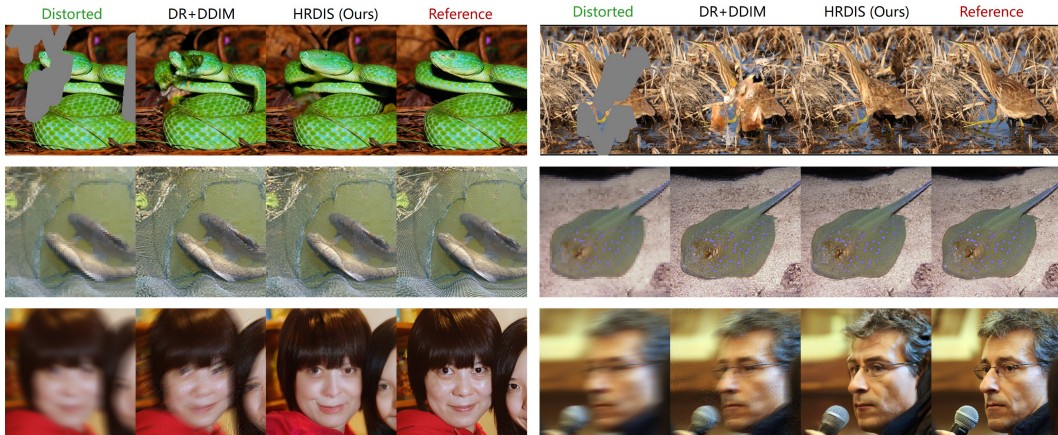

Figure 16: Ablation for combination of DR and DDIM solver.

### C.4 COMPARISON WITH DDS

Here, we show some qualitative comparisons with DDS to demonstrate the advantages of the proposed HRDIS. Among them, Figure 17 shows the evolution of the two methods during the inversion process. Figure 18 shows the results of three linear tasks.

### C.5 ABLATION FOR REGULARIZATION WITH CONSISTENCY MODELS

In this subsection, we present an ablation study, exploring the use of the output from the Consistency Model (CM) (Song et al., 2023b) as a regular bootstrap target for regularization. Due to the lack of available checkpoints for benchmark datasets such as ImageNet or FFHQ at a resolution of 256, we conducted experiments using a checkpoint trained on the LSUN bedroom dataset (Yu et al., 2015). Figure 19 provides qualitative results from these experiments. Our findings indicate that incorporating the CM output as a bootstrap target slightly enhances the restoration of high-frequency details, as it offers a more reliable target for regularization.

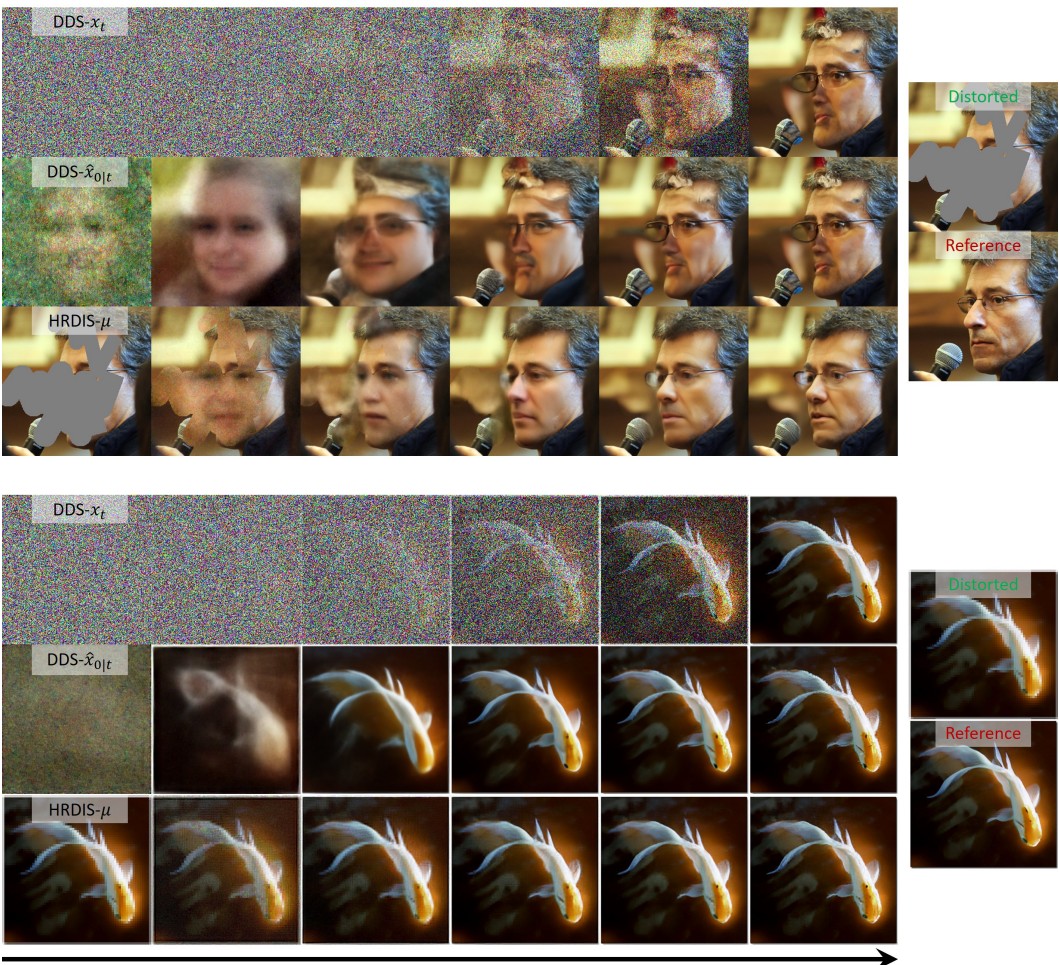

Figure 17: Visualization of evolution in DDS (Chung et al., 2023b) and HRDIS (Ours) for image inversion.

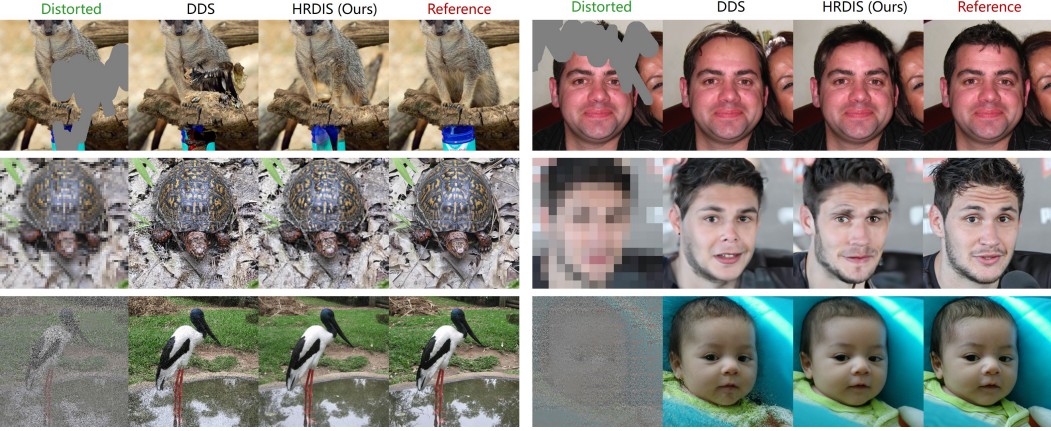

Figure 18: Qualitative comparison with DDS (Chung et al., 2023b).

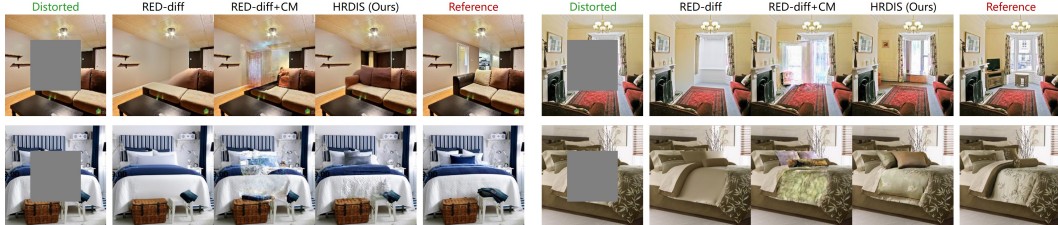

Figure 19: Qualitative results on inpainting using the output of the CM (Song et al., 2023b) as a regularization term.

## C.6 FURTHER EXPERIMENTAL RESULTS

We provide quantitative evaluations based on the standard PSNR and SSIM metrics in Table 9, Table 10 and Table 11.

Table 9: Quantitative evaluation (PSNR, SSIM) of solving *linear* inverse problems on ImageNet 256×256-1k validation dataset. **Bold**: best, underline: second best.

| | Inpaint (10-20%) | | Inpaint (20-30%) | | SR (×4) | | CS (25%) | |
|---|---|---|---|---|---|---|---|---|
| Method | PSNR ↑ | SSIM ↑ | PSNR ↑ | SSIM ↑ | PSNR ↑ | SSIM ↑ | PSNR ↑ | SSIM ↑ |
| DDRM (Kawar et al., 2022) | **26.79** | 0.919 | **23.74** | 0.860 | **26.00** | 0.743 | 24.34 | 0.656 |
| DPS (Chung et al., 2022) | 24.35 | 0.812 | 21.86 | 0.765 | 24.81 | 0.710 | 24.53 | 0.673 |
| ΠGDM (Song et al., 2022) | 22.98 | 0.896 | 20.18 | 0.825 | 25.23 | 0.731 | 28.58 | 0.740 |
| DDNM (Dou & Song, 2024) | 22.58 | 0.894 | 22.54 | 0.865 | 21.39 | 0.529 | 23.39 | 0.588 |
| DDS (Dou & Song, 2024) | 22.02 | 0.807 | 19.26 | 0.815 | 22.25 | 0.580 | 23.02 | 0.569 |
| FPS (Dou & Song, 2024) | 24.81 | 0.845 | 20.76 | 0.793 | 24.90 | 0.703 | 25.77 | 0.690 |
| RED-diff (Mardani et al., 2024) | 26.60 | **0.921** | 23.45 | **0.869** | 25.89 | **0.746** | 26.01 | 0.669 |
| HRDIS (Ours) | 25.65 | 0.913 | 22.12 | 0.835 | 25.54 | 0.708 | **28.79** | **0.807** |

Table 10: Quantitative evaluation (PSNR, SSIM) of solving *linear* inverse problems on FFHQ 256×256-1k validation dataset.

| | Inpaint (30-40%) | | SR (×16) | | CS (10%) | |
|---|---|---|---|---|---|---|
| Method | PSNR ↑ | SSIM↑ | PSNR ↑ | SSIM ↑ | PSNR ↑ | SSIM ↑ |
| DDRM | 23.58 | 0.862 | **22.94** | **0.676** | 24.34 | 0.656 |
| DPS | 22.97 | 0.804 | 20.76 | 0.529 | 25.60 | 0.707 |
| ΠGDM | 21.85 | 0.833 | 21.63 | 0.614 | 27.49 | 0.779 |
| DDNM | 20.43 | 0.816 | 20.83 | 0.574 | 25.13 | 0.708 |
| DDS | 20.07 | 0.810 | 20.77 | 0.570 | 22.39 | 0.609 |
| FPS | 22.17 | 0.828 | 20.95 | 0.553 | 25.86 | 0.715 |
| RED-diff | **23.78** | **0.867** | 22.64 | 0.654 | 26.58 | 0.761 |
| HRDIS (Ours) | 23.06 | 0.837 | 21.75 | 0.622 | **27.85** | **0.802** |

Table 11: Quantitative evaluation (PSNR, SSIM) of solving *nonlinear* inverse problems on FFHQ 256×256-1k validation dataset.

| | Phase retrieval | | HDR | | Nonlinear deblurring | |
|---|---|---|---|---|---|---|
| Method | PSNR ↑ | SSIM↑ | PSNR ↑ | SSIM ↑ | PSNR ↑ | SSIM ↑ |
| DPS | 19.64 | 0.507 | 21.19 | 0.780 | 21.65 | 0.563 |
| RED-diff | 17.97 | 0.488 | 25.97 | 0.869 | 19.59 | 0.484 |
| HRDIS (Ours) | **30.08** | **0.814** | **27.51** | **0.891** | **23.06** | **0.597** |

## C.7 BLIND INVERSE PROBLEM ON REAL-WORLD SAMPLES

In this subsection, we initially explore the utilization of the proposed HRDIS for blind inverse problem-solving on real-world images. To address the challenge of degradation, we conducted

additional experiments by incorporating a learnable degradation model. Inspired by the approach presented in GDP (Fei et al., 2023), we assume a simple degradation model defined as:

$$\mathbf{y} = f\mathbf{x} + \mathcal{M} =: \mathcal{A}_{f,\mathcal{M}}(\mathbf{x}),$$

where $f$ is a scalar and $\mathcal{M}$ is a mask, both of which are initially unknown. These parameters, along with the image $x$, are optimized alternately using HRDIS. The optimization is performed as follows:
1. Update $f$ and $\mathcal{M}$: These parameters are updated using the gradient:

$$\nabla_{f,\mathcal{M}}\|\mathbf{y} - \mathcal{A}_{f,\mathcal{M}}(\mu)\|^2.$$

2. Update $\mu$: The image is updated using the gradient:

$$\nabla_{\mu}\|\mathbf{y} - \mathcal{A}_{f,\mathcal{M}}(\mu)\|^2 + \mathbb{E}[\omega_t(\boldsymbol{\epsilon}_\theta(\mu_t, t) - \boldsymbol{\epsilon}_{\text{hybrid}})].$$

We performed preliminary experiments on real-world low-light images from the LOL dataset (Wei et al., 2018). The results in Table 12 demonstrate that HRDIS achieves superior performance compared to GDP-$x_t/x_0$ (Fei et al., 2023), with the added advantage of reduced computational cost. Specifically, HRDIS requires only about 300 NFE, compared to 1000 NFE for GDP. Figure 20 provides qualitative results that illustrate the effectiveness of HRDIS in reconstructing high-quality images from degraded inputs. These findings highlight HRDIS's potential in addressing blind inverse problems effectively and efficiently.

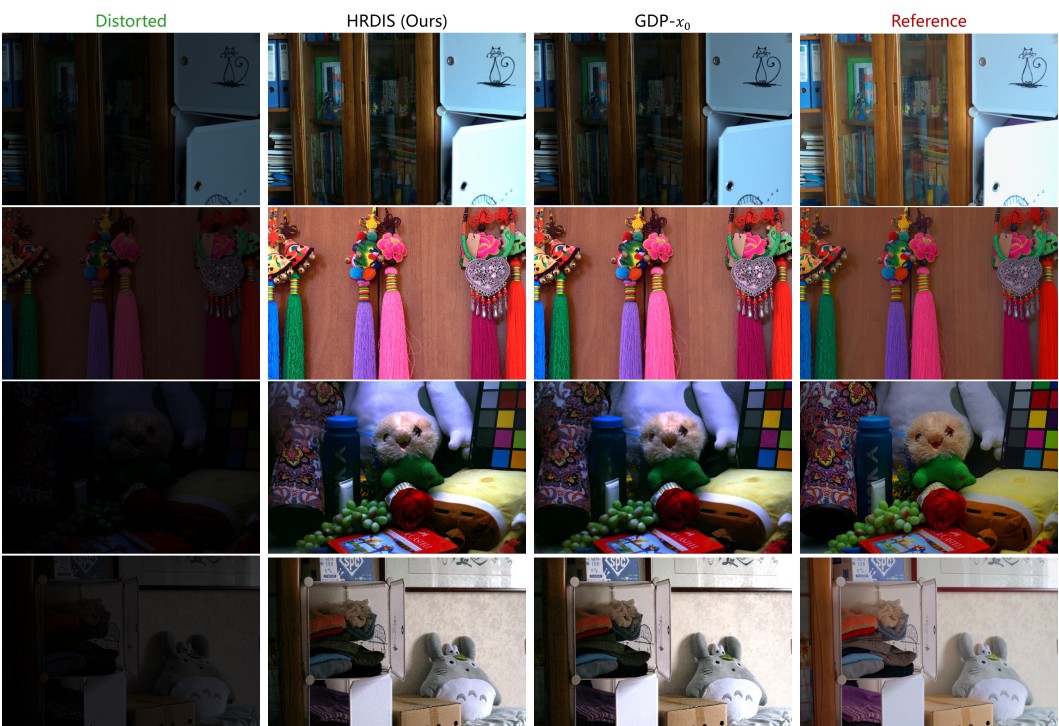

Figure 20: Qualitative comparison of blind reversal problems.

Table 12: Quantitative evaluation of blind inverse problem on the LOL dataset.

| Method | PSNR $\uparrow$ | SSIM $\uparrow$ | FID $\downarrow$ |
|---|---|---|---|
| GDP-$x_t$ | 7.32 | 0.57 | 238.92 |
| GDP-$x_0$ | 13.93 | 0.63 | 75.16 |
| HRDIS (Ours) | **14.25** | **0.65** | **74.89** |

## C.8 ABLATION OF $\varepsilon$

To investigate the influence of $\varepsilon$ in the CR term, we conducted an ablation study focusing on inpainting and super-resolution tasks using the FFHQ dataset. The diffusion model operates over a normalized time interval $[0, 1]$, and we evaluated the performance of $\varepsilon$ across several magnitudes.

The quantitative results of the ablation study are presented in Figure 21. We observed that $\varepsilon$ performs optimally when it is around the order of $10^{-2}$. Key observations include:

- When $\varepsilon$ is too small ($10^{-3}$), the CR effect diminishes, resulting in blurred outputs.
- When $\varepsilon$ is too large ($10^{-1}$), the resulting image quality degrades due to the discretization error.

These findings highlight the importance of carefully selecting $\varepsilon$ to achieve optimal performance. Figure 22 provides qualitative visualizations of the ablation study. These examples clearly demonstrate the impact of different $\varepsilon$ values on the final output.

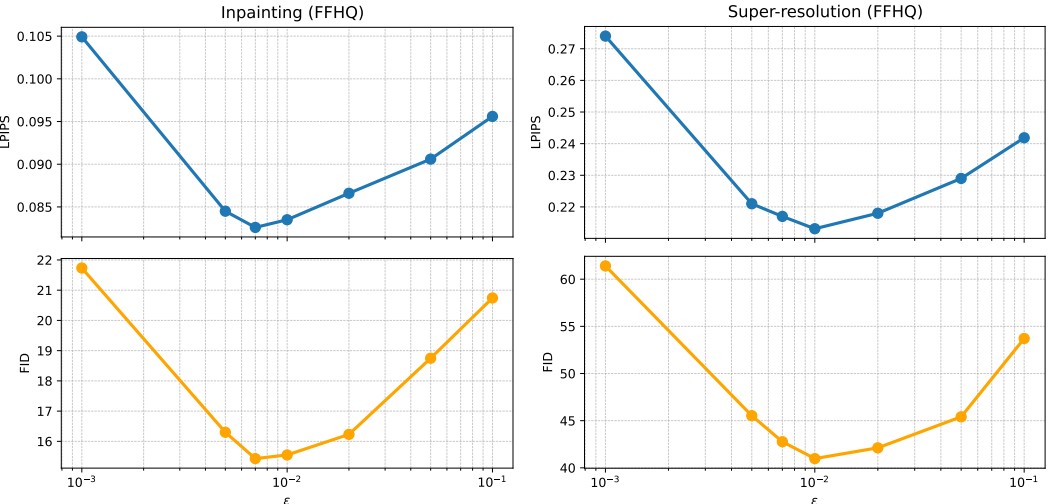

Figure 21: Quantitative results for varying values of $\varepsilon \in \{0.001, 0.005, 0.007, 0.01, 0.02, 0.05, 0.1\}$.

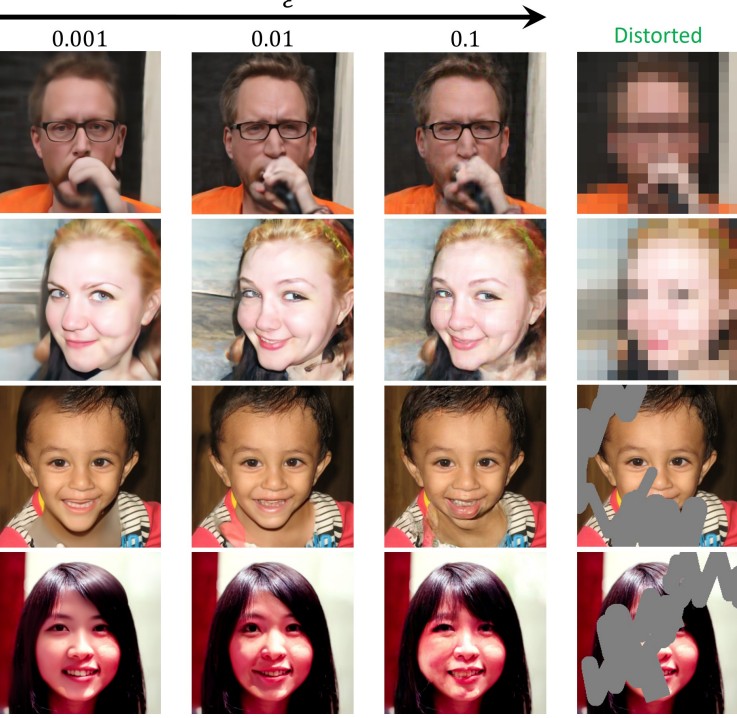

Figure 22: Visualization of the output of different $\varepsilon$.

## C.9 ADDITIONAL FIGURES

In this subsection, we show additional qualitative results for HRDIS.

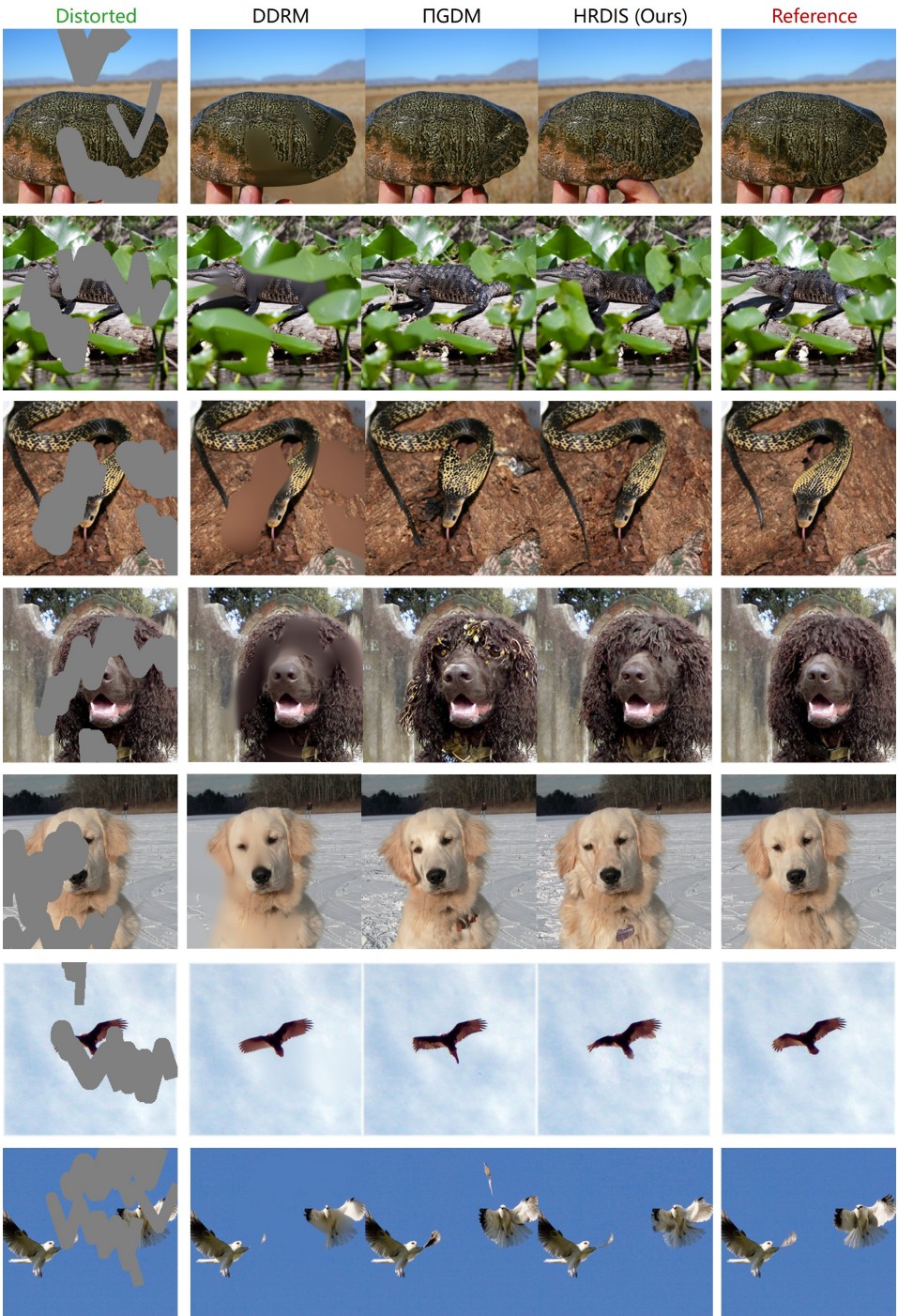

Figure 23: Comparing methods for the image inpainting problem on ImageNet $256 \times 256$, including DDRM (Kawar et al., 2022), ΠGDM (Song et al., 2022) and HRDIS (Ours).

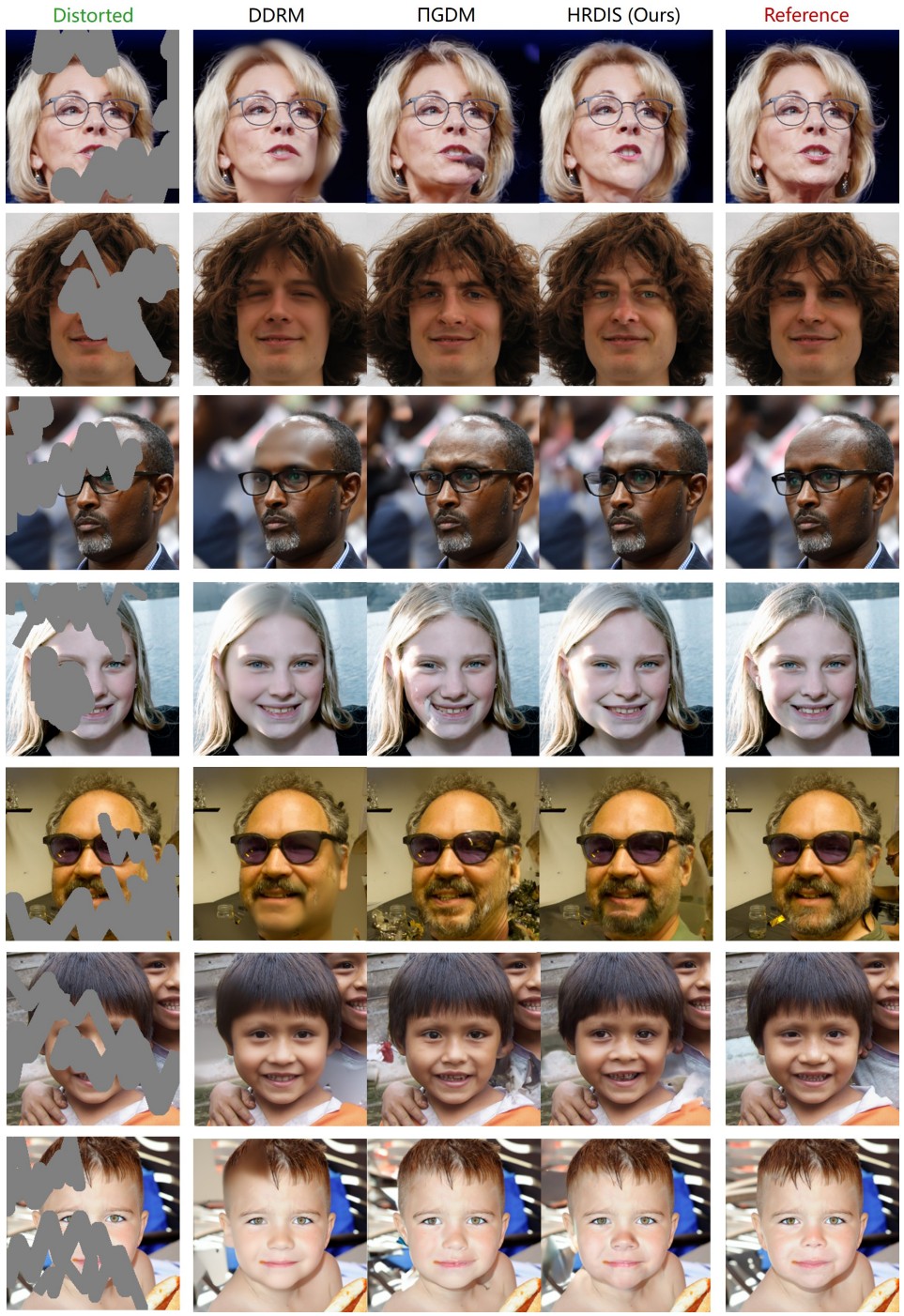

Figure 24: Comparing methods for the image inpainting problem on FFHQ $256 \times 256$, including DDRM (Kawar et al., 2022), ΠGDM (Song et al., 2022) and HRDIS (Ours).

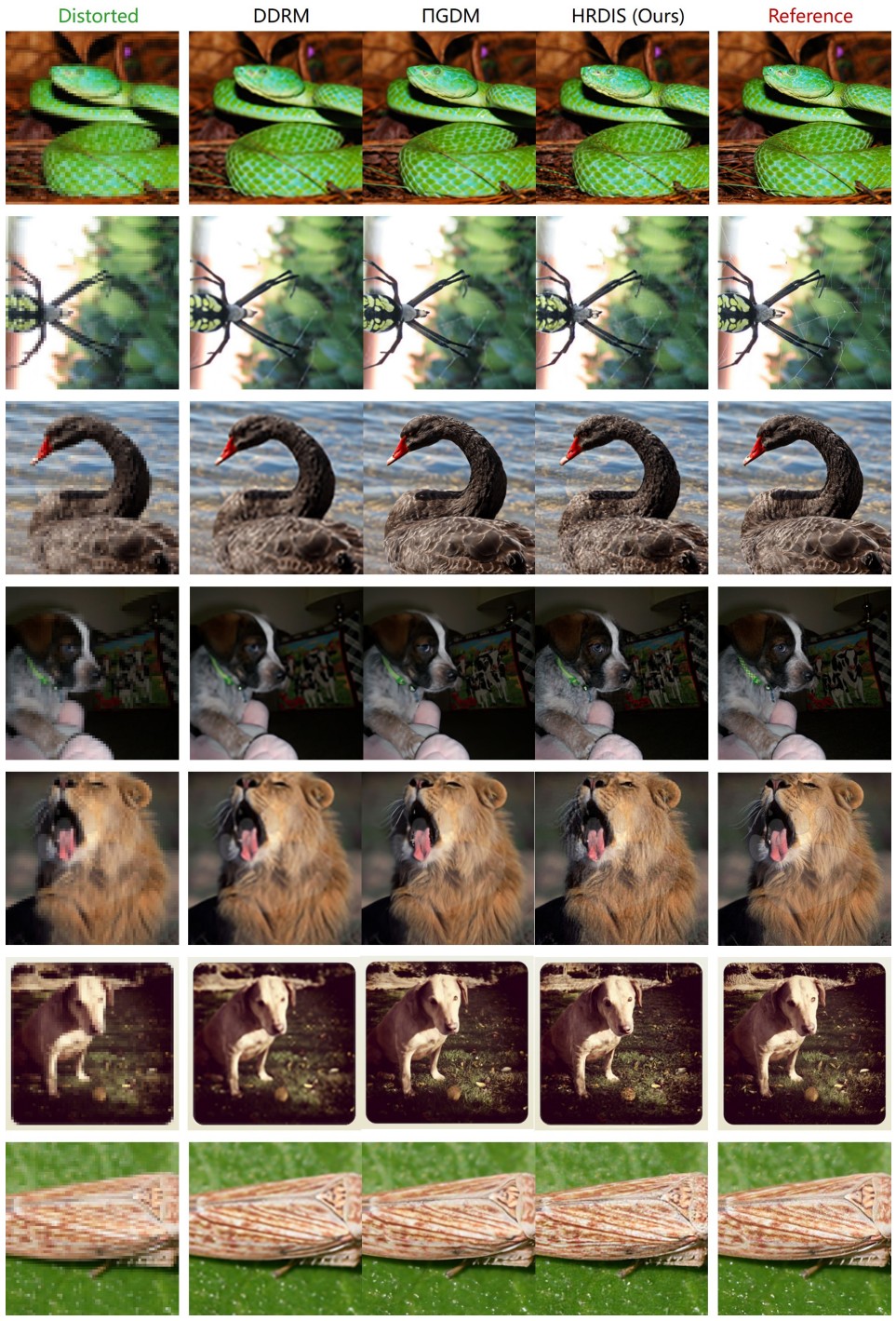

Figure 25: Comparing methods for the image super-resolution problem on ImageNet $256 \times 256$, including DDRM (Kawar et al., 2022), ΠGDM (Song et al., 2022) and HRDIS (Ours).

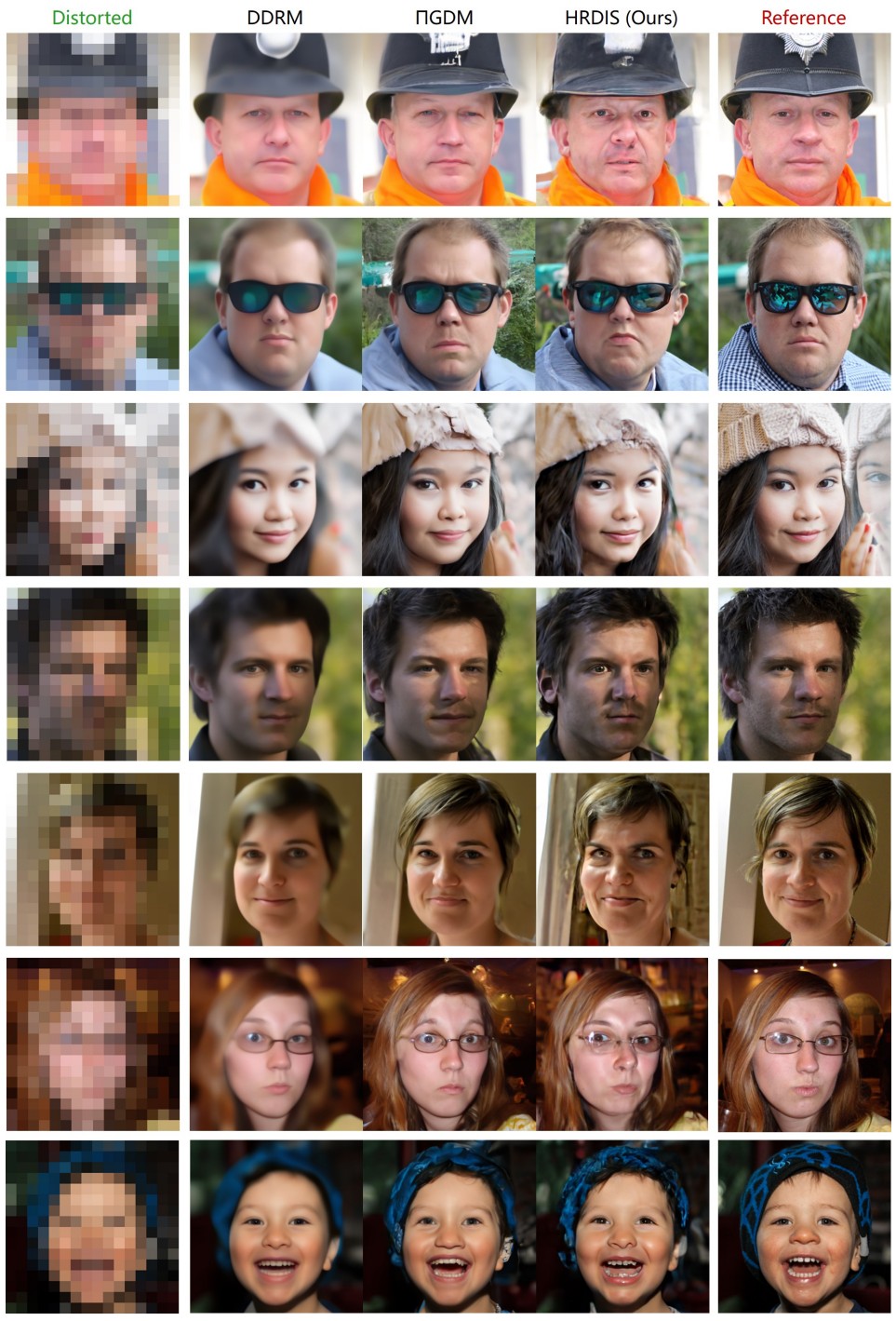

Figure 26: Comparing methods for the image super-resolution problem on FFHQ $256 \times 256$, including DDRM (Kawar et al., 2022), ΠGDM (Song et al., 2022) and HRDIS (Ours).

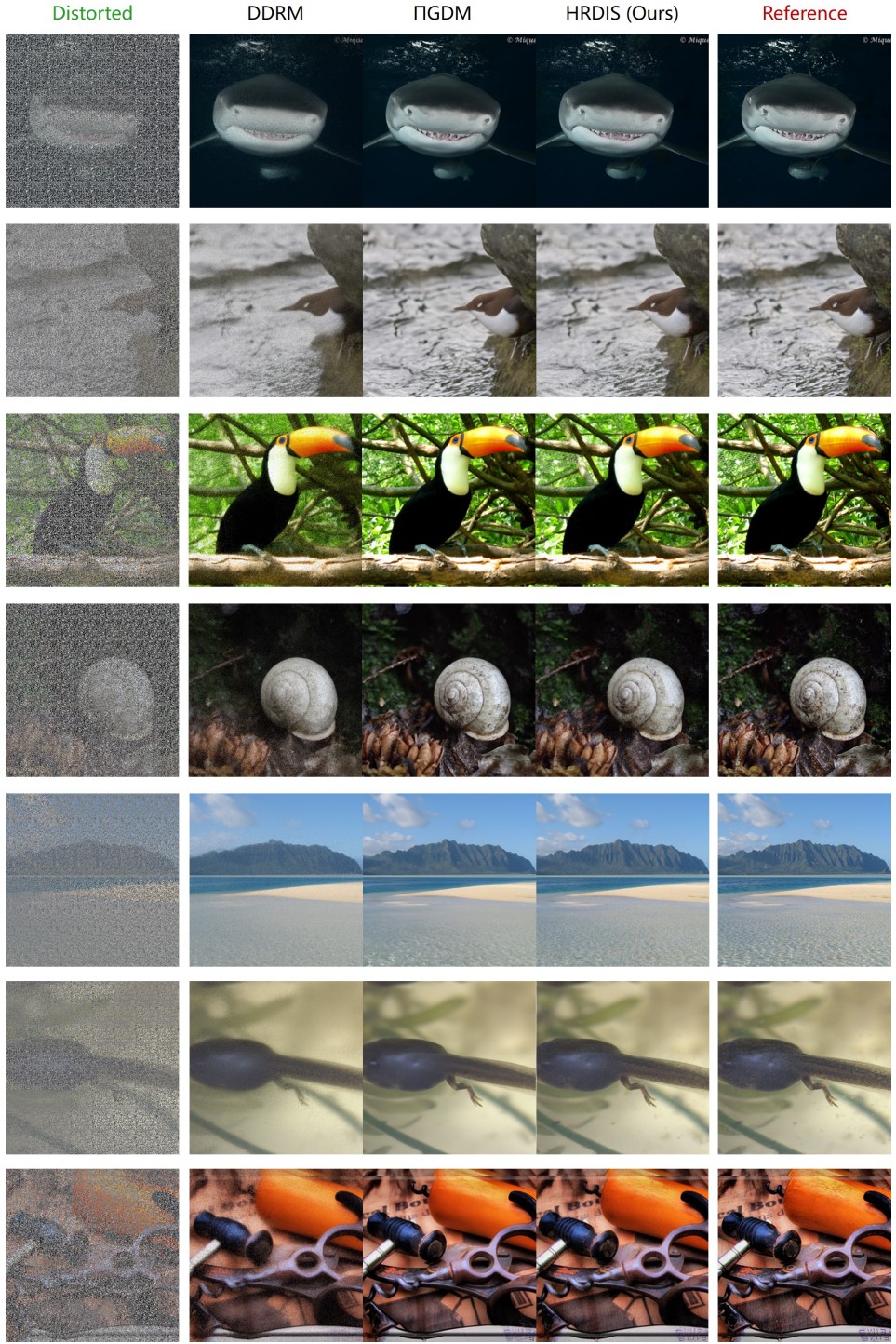

Figure 27: Comparing methods for the compressed sensing problem on ImageNet $256 \times 256$, including DDRM (Kawar et al., 2022), ΠGDM (Song et al., 2022) and HRDIS (Ours).

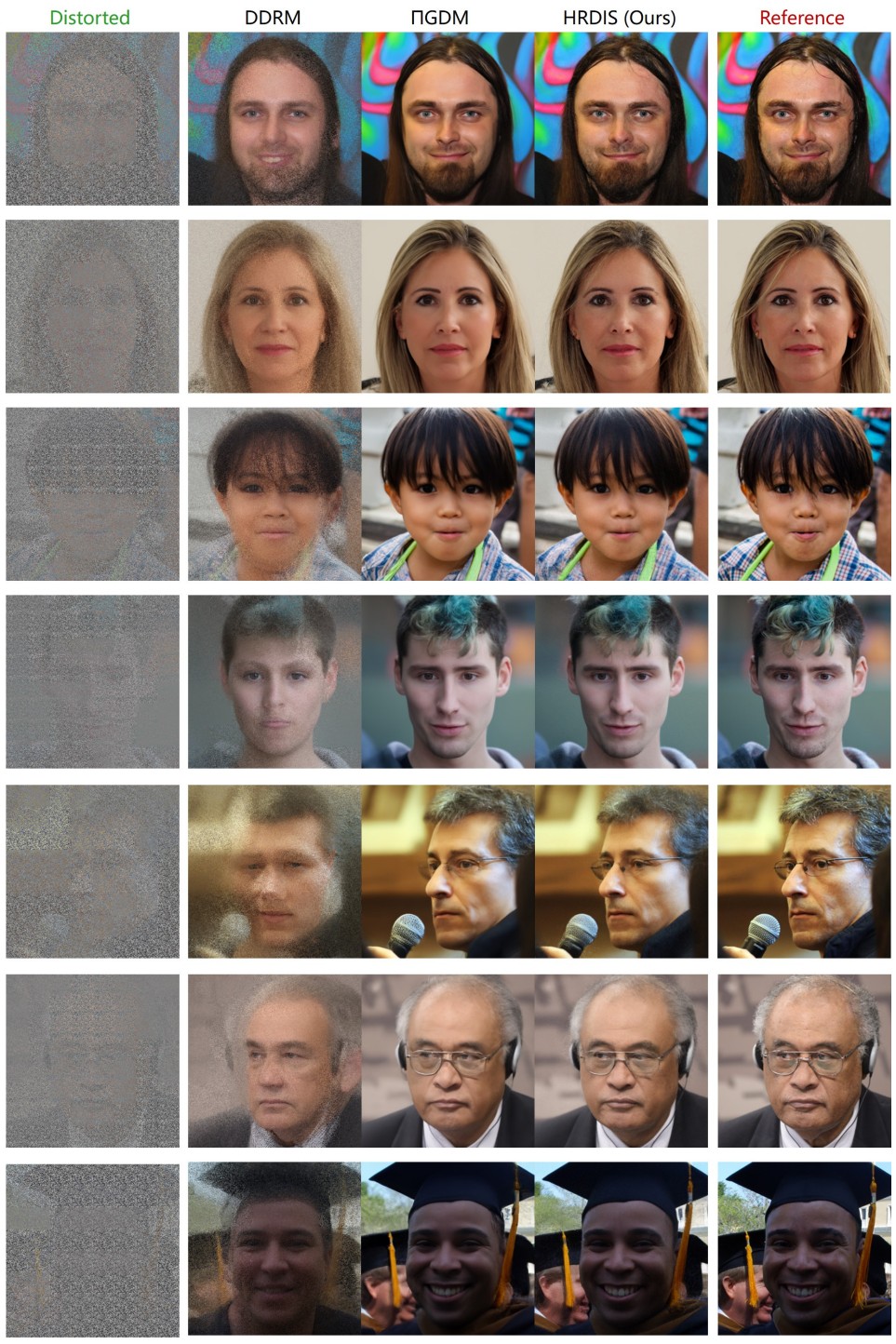

Figure 28: Comparing methods for the compressed sensing problem on FFHQ $256 \times 256$, including DDRM (Kawar et al., 2022), ΠGDM (Song et al., 2022) and HRDIS (Ours).

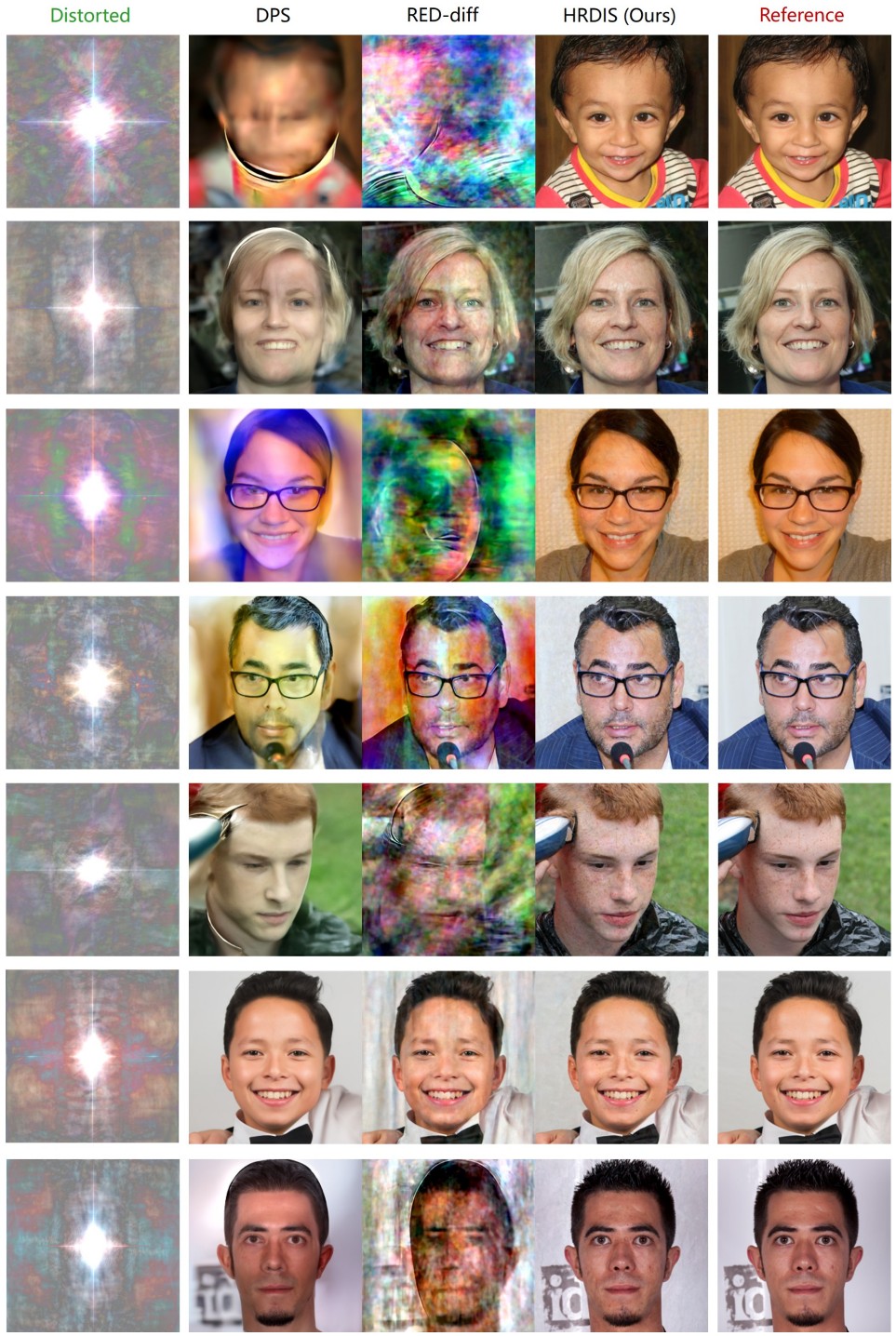

Figure 29: Comparing methods for the phase retrieval problem on FFHQ $256 \times 256$, including DPS (Chung et al., 2022), RED-diff (Mardani et al., 2024) and HRDIS (Ours).

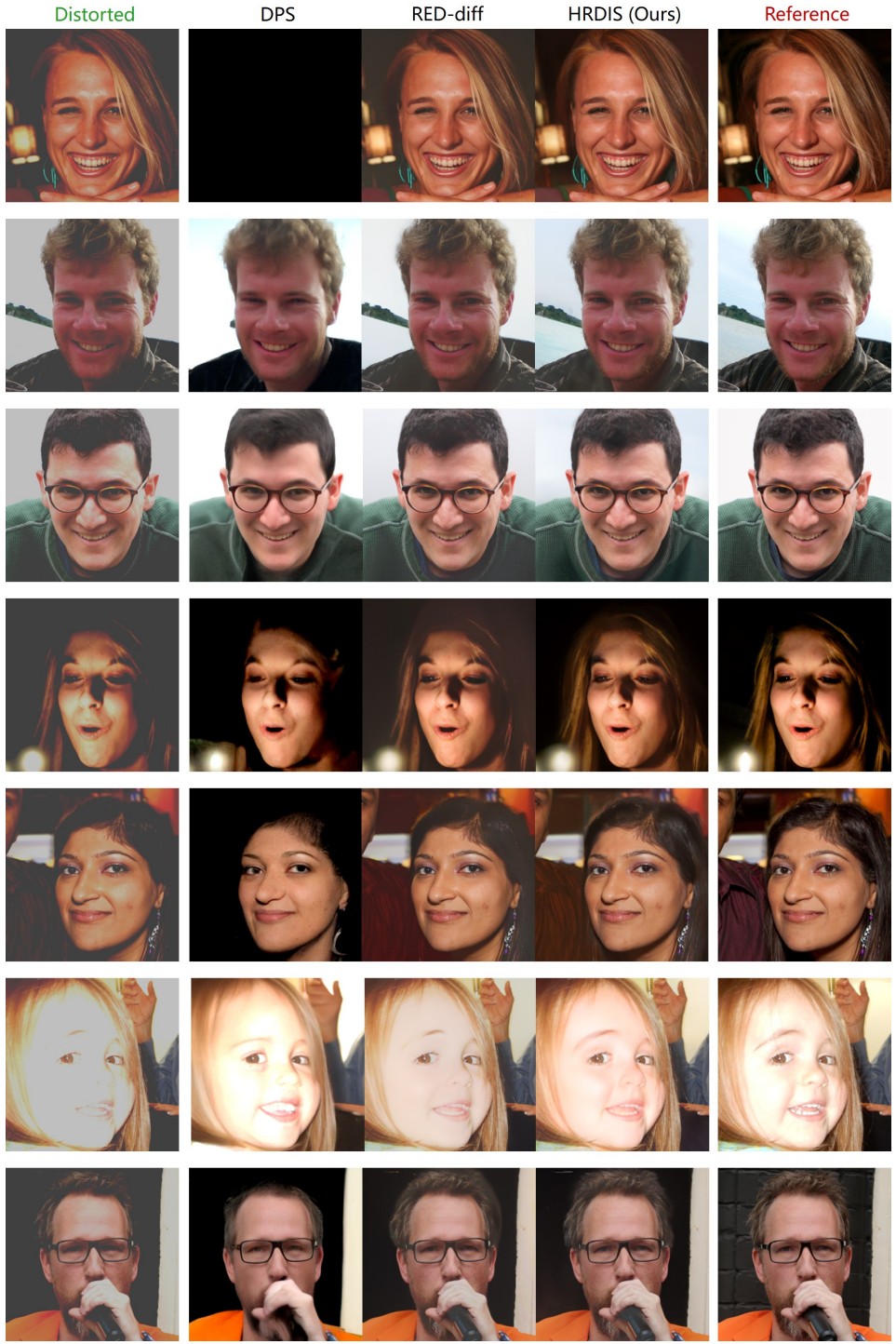

Figure 30: Comparing methods for the nonlinear HDR problem on FFHQ $256 \times 256$, including DPS (Chung et al., 2022), RED-diff (Mardani et al., 2024) and HRDIS (Ours).

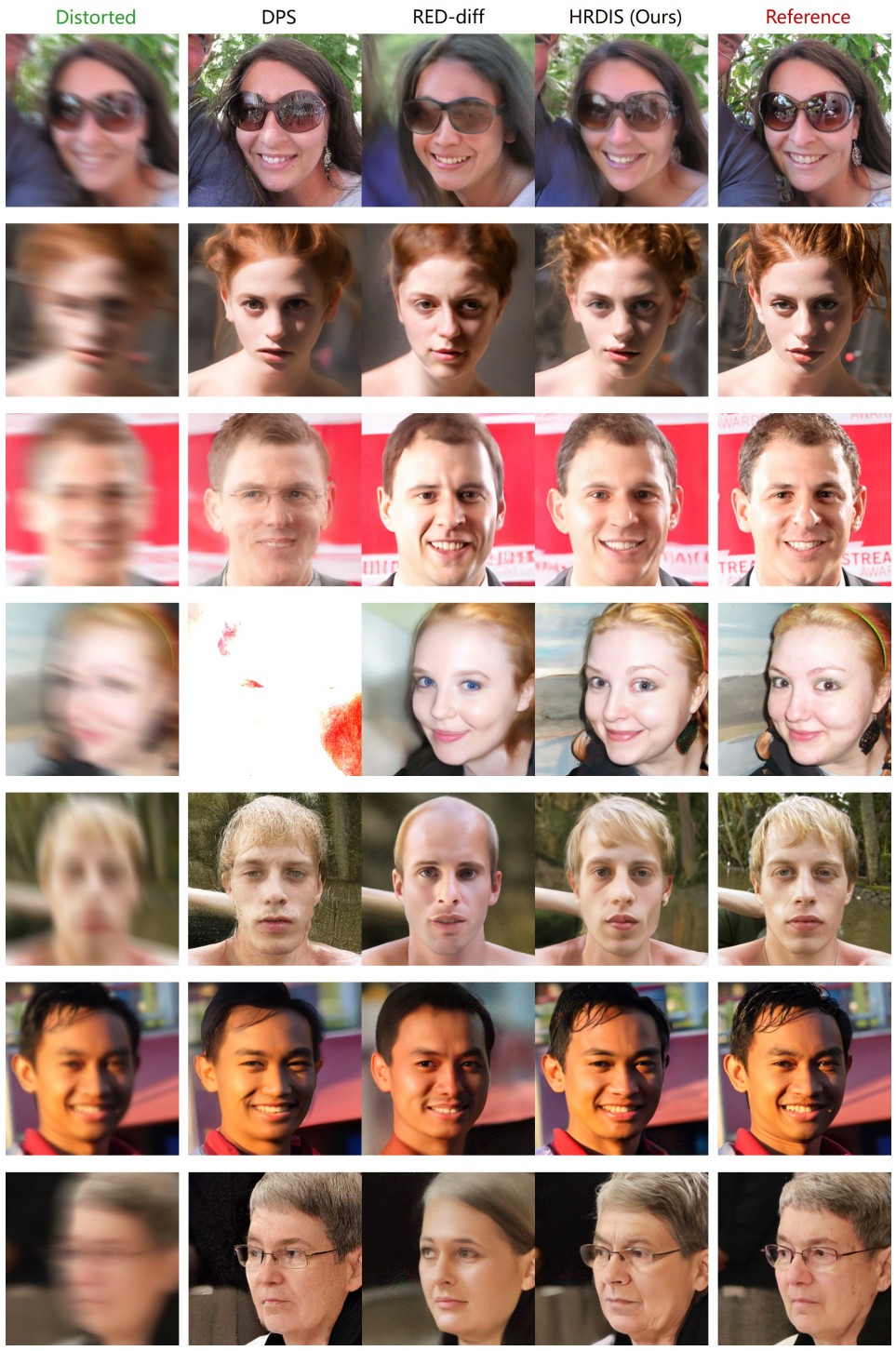

Figure 31: Comparing methods for the nonlinear deblurring problem on FFHQ $256 \times 256$, including DPS (Chung et al., 2022), RED-diff (Mardani et al., 2024) and HRDIS (Ours).

