# OpenReview forum: "Hybrid Regularization Improves Diffusion-based Inverse Problem Solving"
_ICLR.cc/2025/Conference — ICLR 2025 Poster_

### Official Review · Reviewer_86ej · 2024-10-20

**Soundness:** 3
**Presentation:** 3
**Contribution:** 3
**Rating:** 6
**Confidence:** 3

**Summary:**

This paper proposes a novel approach called Consistency Regularization (CR), which provides stabilized gradients without the need for ODE simulations. Building on this, we introduce Hybrid Regularization (HR), a unified framework that combines the strengths of both DR and CR, harnessing their synergistic potential. The approach proves to be effective across a broad spectrum of inverse problems, encompassing both linear and nonlinear scenarios, as well as various measurement noise statistics. The experiment is clear and convincing.

**Strengths:**

1. The method combines the strength of both Consistency Regularization and Denoising processes with Regularization.
2. The experimental results are promising, including non-linear problems.
3. The writing quality is strong.

**Weaknesses:**

1. The method lacks comparisons with recent works such as DDNM [a] and GDP [b].
2. Although the method addresses non-linear problems, it has not been evaluated on blind problems. The performance on real-world samples should be demonstrated to validate its practical effectiveness.


[a] Zero-Shot Image Restoration Using Denoising Diffusion Null-Space Model, ICLR 2023.
[b] Generative Diffusion Prior for Unified Image Restoration and Enhancement, CVPR 2023.

**Questions:**

1. It would be interesting to explore whether the method proposed in this paper can be extended to other inversion problems, such as image editing.

---

> ### Author Response · Authors · 2024-11-23
> **Responses to Reviewer 86ej**
>
> Dear Reviewer 86ej，
>
> Thank you for your time and constructive comments on our work. Below are the responses we have prepared to your questions.
>
> ***
> **Q: Comparisons with recent works such as DDNM [1] and GDP [2].**
>
> **A:** Thank you for your advice. We have incorporated the methods you mentioned into our experiments. The implementations of DDNM and GDP are from official repositories. We have performed inpainting, super-resolution, and compressed sensing experiments on ImageNet and FFHQ. The results are shown below, while **Tables 1 and 2 (Page 8)** of the manuscript have been updated.
>
> |Task|        | LPIPS ↓ (ImageNet) | FID ↓ (ImageNet) | CA ↑ (ImageNet) | LPIPS ↓ (FFHQ) | FID ↓ (FFHQ) |
> |----|------------|--------------------|------------------|------------|----------------|--------------|
> || DDNM  | 0.106 | 26.61 | 67.9  | 0.103 | 19.72 |
> || GDP  | 0.135 | 27.83 | 67.3  | 0.112 | 24.38 |
> | Inpainting| HRDIS (Ours) |  **0.096**   |  **20.10**   |  **71.3** |    **0.082**  |  **15.42** |
> || DDNM          | 0.251    | 47.15 | 61.9      | **0.210**   | 41.50           |
> || GDP          | 0.234    | 44.08 | 62.4      | 0.237   | 43.51           |
> |SR| HRDIS (Ours) |  **0.137**   | **33.01** | **68.8**      | 0.213    | **40.97**       |
> || DDNM          | 0.247     | 53.16 | 58.4     | 0.154    | 56.15           |
> || GDP         | 0.283|  60.01 | 52.3     | 0.267 | 91.84           |
> |CS| HRDIS (Ours) | **0.059**  | **22.04**   | **72.3**     | **0.088**   | **35.27**          |
>
>
> As can be seen from the results, HRDIS is consistently superior to these advanced methods.
>
> ***
> **Q: The performance on real-world samples of blind inverse problems**
> ﻿
>
> **A:** A known limitation of our approach is the requirement to access the degradation model, which is a common challenge for all training-free diffusion-based inverse problem solvers. We have frankly recognized this limitation in **Appendix A.2 (Page 14)** of the manuscript.
> ﻿
> Be that as it may, we conducted additional experiments for blind inverse problems. One potential way to mitigate the issue is to assume a simple, learnable degradation model and jointly optimize it. Following the approach in [2], we consider the degradation model as:
> ﻿
> $y=fx+\mathcal { M }=:A_{f,\mathcal { M }}(x)$
> ﻿
>
> where $f$ is a scalar and $\mathcal { M }$ is a mask, both of which are initially unknown. We then alternately optimize these parameters and the image using HRDIS through the following steps:
> ﻿
> 1. **Update $f$ and $\mathcal { M }$:** Optimize them using the gradient $\nabla_{f,\mathcal { M }}\\|y-A_{f,\mathcal { M }}(\mu)\\|^2$.
> 2. **Update $\mu$:** Optimize the image using the gradient $\nabla_{\mu}\\|y-A_{f,\mathcal { M }}(\mu)\\|^2+\mathbb{E}[\omega_t (\epsilon_{\theta}- \epsilon_{hybrid})]$.
> ﻿
> We performed preliminary experiments on real-world low-light data (LOL dataset). Our results demonstrate that HRDIS achieves performance superior to GDP (**Table 12, Page 24**), with the added advantage of being faster. Specifically, HRDIS requires about 300 NFE, compared to GDP's 1000 NFE. **Figure 19 (Page 24)** in the revised manuscript provides qualitative results for the reviewer's reference.
>
> ***
> **Q: Extention to other inversion problems, such as image editing.**
>
> **A:** To be honest we have limited knowledge of recent advances in image editing. It seems to us that our proposed approach has the potential to be extended to this area. For instance, our inpainting function could be adapted to modify specific regions of an image with a mask. Additionally,  it is possible to combine our hybrid regularization with techniques like Delta denoising score [3]. We acknowledge this as an exciting avenue for future exploration and plan to investigate it further in our subsequent work.
> ***
> [1] *Zero-Shot Image Restoration Using Denoising Diffusion Null-Space Model, ICLR 2023.*
>
> [2] *Generative Diffusion Prior for Unified Image Restoration and Enhancement, CVPR 2023.*
>
> [3] *Hertz A, Aberman K, Cohen-Or D. Delta denoising score. ICCV 2023.*

---

> > ### Comment · Reviewer_86ej · 2024-11-25
> >
> > I would like to thank the authors for thoroughly addressing all my concerns, updating the paper, and providing excellent experimental results. I will review the feedback from other reviewers before finalizing my score update.

---

> > > ### Author Response · Authors · 2024-11-28
> > >
> > > Dear Reviewer 86ej，
> > >
> > > Thank you for your thoughtful consideration. We are pleased to hear that our response has addressed your concerns, and we look forward to your final recommendation.
> > >
> > > Best,
> > >
> > > Authors of Submission 9721

---

### Official Review · Reviewer_HwgS · 2024-10-26

**Soundness:** 4
**Presentation:** 4
**Contribution:** 3
**Rating:** 6
**Confidence:** 4

**Summary:**

This paper is an improvement over previous work called RED-diff, which is variational method for diffusion posterior estimation.

By analyzing the weakness of RED-diff and drawing inspiration from consistency model, this paper proposes consistency regularization to replace the denoising regularization in RED-diff.

By analyzing the limitation of consistency regularization and drawing inspiration from DDIM, this paper further proposes hybrid regularization which combines denosing and consistency.

Experiments show the proposed method achieve great improvement regarding image quality and sampling speed.

**Strengths:**

The writing follows a problem analysis and problem soving manner, which is very logical and frendly to readers.

The improvement over previous works regarding image quality and speed is significant. The experiments are thorough, especially, different previous works are analyzed in detail regarding their image quality.

The Consistency Regularization (CR)  and Hybrid Regularization (HR) are all well motivated by previous works on diffusion sampling, i.e., consistency model and DDIM etc.

**Weaknesses:**

eq. 10 is already derived in the previous work Mardani et al. (2024) in section 4.3, it might be better to clarify this.

There exists a previous work which uses consistency model for diffusion posterior sampling ('Consistency Models Improve Diffusion Inverse Solvers') and should be discussed. What if we follow this paper to use consistency model directly in the original RED-diff framework?  Can this, to some extent, play the same role as Consistency Regularization (CR)?

[main concern] The consistency to the measurement is not evaluated and compared. This line of works solve inverse problems, shoud the consistency to the measurement also be a important metric in addition to the image reconstruction quality? For example, In DPS and RED-diff, PSNR between reconstruction and ground truth is reported, which can represent the consistency to measurement to some extent. I am not requiring achieving SOTA PSNR and SOTA FID at the same time. But it helps to have some fidelity evaluation.

**Questions:**

It is interesting that beta=0.2 is a common sweet spot for across different tasks, is there any insight on this?

---

> ### Author Response · Authors · 2024-11-22
> **Responses to Reviewer HwgS**
>
> Dear HwgS，
>
> Thank you for your constructive feedback. Below is our detailed response to your concerns.
>
> ***
> **Q: Clarify for eq. 10.**
>
> **A:** Thank you for your suggestion, we have clarified this in the revised manuscript.
>
> ***
> **Q: Discussion on 'Consistency Models Improve Diffusion Inverse Solvers'[1].**
>
> **A:**  The work in [1] focuses on leveraging consistency models (CMs) to enhance diffusion posterior sampling (DPS) [2]. In DPS, the single-step denoising step represents the mean of clean samples, while the single-step estimation in CMs ensures the prediction of clean samples. However, a limitation of this approach is that it depends on **pre-trained CMs, which are not widely available**. In contrast, our method does not face this limitation. We utilize a diffusion model, which has the advantage of richer pre-training checkpoints that are more generally accessible.
>
> ***
> **Q: Use CMs directly in the original RED-diff framework.**
>
> **A:** Unfortunately the repository for CMs (https://github.com/openai/consistency_models) does not provide checkpoints for the benchmark datasets ImageNet256 or FFHQ256. We decided to experiment with a checkpoint on the LSUN bedroom. **Figure 18 (Page 23)** of the revised manuscript shows some qualitative results on box inpainting. We find that using a CM slightly improves the RED-diff of high-frequency features,  which is similar to the role of CR to a certain extent.
>
> ***
> **Q:  Report on reconstruction metrics.**
>
> **A:**  In the revised manuscript, we have included additional reporting on PSNR and SSIM metrics on **Tables 9-11 (Page 23)**. Our findings indicate that HRDIS performs strongly on compressed sensing and three nonlinear problems. For scenarios involving free-form masks or large-scale downsampling, uncertainty increases, resulting in a trade-off between PSNR and FID. Methods such as DDRM and RED-diff, which often produce feature-averaged reconstructions, tend to achieve higher PSNR scores. HRDIS prioritizes generating perceptually consistent reconstructions.
>
> ***
> **Q: $\beta$=0.2 is a common sweet spot for different tasks.**
>
> **A:** In fact, we performed the ablation on $\beta$ over a relatively wide range, evaluating the values 0.0, 0.1,0.2,0.5,1.0 within the interval [0,1]. We believe that if finer intervals are taken, the optimal $\beta$ should be slightly different for each task. Nevertheless, a uniform setting of $\beta=0.2$ is enough for our framework to compete with the SOTA method.
>
> ***
> [1] *Xu, Tongda, et al. Consistency Models Improve Diffusion Inverse Solvers. arXiv preprint arXiv:2403.12063 (2024).*
>
> [2] *Chung, Hyungjin, et al. Diffusion Posterior Sampling for General Noisy Inverse Problems. ICLR 2023.*

---

> > ### Comment · Reviewer_HwgS · 2024-11-24
> >
> > Thanks for addressing my concerns. I am considering other reviewers' comments carefully before the final score recommendation.

---

> > > ### Author Response · Authors · 2024-11-28
> > >
> > > Dear Reviewer HwgS，
> > >
> > > Thank you for your thoughtful consideration and for reviewing our rebuttal. We greatly appreciate your careful evaluation of our work and look forward to your final recommendation.
> > >
> > > Best,
> > >
> > > Authors of Submission 9721

---

### Official Review · Reviewer_tzUN · 2024-10-28

**Soundness:** 3
**Presentation:** 3
**Contribution:** 3
**Rating:** 6
**Confidence:** 3

**Summary:**

This paper proposes a simple and effective hybrid regularization scheme for solving inverse problems using pretrained diffusion models. The authors first analyze the potential issues of existing optimization-based inverse problem solving methods, such as Denoising Regularization (DR). Based on the analysis, they propose Consistency Regularization (CR) to address the inaccurate gradient issues inspired by Consistency Distillation (CD). Then, a Hybrid Regularization (HR) method is further proposed to benefit from both the DR and CR. The proposed method is evaluated on FFHQ and ImageNet datasets, with comprehensive experiments on different distortion and inverse problems. The results show that the proposed method is efficient and can reconstruct better details than the state-of-the-art methods.

**Strengths:**

- The proposed Consistency Regularization (CR) and Hybrid Regularization (HR) methods are simple and effective, and can be applied to both linear and nonlinear inverse problems.
- The results look much sharper than RED-diff (e.g., Figure 1) thanks to the proposed CR and HR methods. Also, the results are overall outperforming the state-of-the-art methods.
- The reference seems comprehensive. The authors even mention score distillation sampling (SDS) introduced in text-to-3D literature, which indeed shares some similarity with RED-diff and the proposed method.
- The paper includes comprehensive results, comparisons and ablation studies that are important to understand the effect of the proposed method.
- It is nice that the authors provide pseudo code in the appendix and code in the supplemental.

**Weaknesses:**

- The proposed CR method can lead to over-sharpened or noisy images, but it is not clear to me why this happens. To my understanding, this is not the case for image generation from noise based on the original consistency distillation paper [Song et al. 2023]. As shown in Figure 2 and Figure 3, the Consistency Regularization (CR) results do get better structures but also introduce artifacts and noise. Also in Figure 1 middle column, the images look a bit noisy even under the Hybrid Regularization (HR) method. In Figure 10 the problem seems more visible ($\beta=0$). I would like to hear more insights on this if this is a potential issue.

**Questions:**

- Would the idea of Consistency Regularization (CR) lead to less diversity in the inversed image? As shown in the original consistency distillation paper [Song et al. 2023], the amount of variations in the generated images  (measured by recall [Kynka¨anniemi et al. 2019]) can be reduced due to consistency distillation.
- As discussed in the "Weakness" section above, could the authors provide more insight on why the proposed method can lead to over-sharpened or noisy images?
- The equation of consistency distillation has a moving average (EMA) term in the original paper [Song et al. 2023], denoted by $\theta^{-}$. Is it also the case in Equation 12 or the same network is applied for two different time steps?
- Equation 14 shows that the regularization requires 2x network inference denoted by epsilon_theta, but this does not slow down the optimization process. Is it because the effect of reducing the number of iterations is more significant than the extra computation cost?
- I am a bit confused about what should I expect in Figure 6, since I am not familiar with compressed sensing and phase retrievel. Given such highly distorted images, is it possible to recover the results that can match the reference so well? The same question applies to Figure 7 on the nonlinear deblurring results.

Some minor comments:
- In line 421-422, "FFHD" should be "FFHQ".

---

> ### Author Response · Authors · 2024-11-22
> **Responses to Reviewer tzUN**
>
> Dear Reviewer tzUN,
>
> Thank you for your valuable feedback and thoughtful comments on our manuscript. We will address your questions point by point below.
>
> ***
> **Q:  Further insights for over-sharpness or noise in Consistency Regularization (CR)**
>
> **A:** We hypothesize that two aspects cause the results of CR. On the one hand, CR directly optimizes image pixels, and consistency distillation (CD) is equivalent to optimizing a neural representation of the image. It is well known that neural representation naturally has implicit regularization [1,2], whose spectral bias is resistance to over-sharpness and noise. On the other hand, CR optimization may be more challenging because it also involves measurement loss compared to CD.
>
> ***
> **Q: Less diversity in the inversed image.**
>
> **A:** Experimentally we have not observed a reduction in diversity so far. In fact, HRDIS mitigates the mode collapse of the original RED-diff, allowing the use of different seeds to produce diverse results.
>
> ***
> **Q: The use of EMA.**
>
> **A:** CD did initially use the EMA (Exponential Moving Average) technique. However, this trick was abandoned in subsequent improvements by Song et al. [3]. As a result, we do not use EMA in our method either.
>
> ***
> **Q: 2x network inference.**
>
> **A:** In practice, due to the use of the descending timestep strategy proposed in RED-diff, we suggest retaining the predicted noise from the previous step to approximate the difference in CR. This results in almost no increase in computational cost for each optimization step compared to RED-diff. At the same time, HRDIS usually reduces the number of iterations because the gradient is more stable.
>
> ***
> **Q: The results in Figures 6 and 7.**
>
> **A:** The degradation setups used for our experiments mainly align with those adopted in prior works.
> ﻿
>
> Phase retrieval involves recovering the phase information in the Fourier domain [4], which is inherently unstable. We followed the setup of [4] and took the best one of four independent runs for each method.
>
> Nonlinear deblurring simulates the blurring kernel using a pre-trained UNet model to replicate realistic degradation scenarios [4].
>
> For compressed sensing, we utilize block-based sampling, where the signal is decomposed using Singular Value Decomposition (SVD) to extract its primary sparse features. This operation is performed in blocks, each of size 32 × 32 [5].
> ﻿
>
> HRDIS usually has superior performance on these tasks, consistently recovering higher-quality results compared to the baseline methods.
>
> ***
> **Q: In line 421-422, "FFHD" should be "FFHQ"**
>
> **A:** Thanks for your careful reading. We have corrected the clerical error.
>
> ***
> [1] *Ulyanov, Dmitry, Andrea Vedaldi, and Victor Lempitsky. Deep image prior. CVPR 2018.*
>
> [2] *Rahaman, Nasim, et al. On the spectral bias of neural networks. ICML 2019.*
>
> [3] *Song, Yang, and Prafulla Dhariwal. Improved techniques for training consistency models. ICLR 2024.*
>
> [4] *Chung H, Kim J, Mccann M T, et al. Diffusion Posterior Sampling for General Noisy Inverse Problems. ICLR 2023.*
>
> [4] *Wang Y, Yu J, Zhang J. Zero-Shot Image Restoration Using Denoising Diffusion Null-Space Model. ICLR 2023.*

---

> > ### Comment · Reviewer_tzUN · 2024-11-24
> >
> > I would like to thank the authors for clarifications and updating the paper.
> >
> > - For the over-sharpness/noisy results, thanks for pointing out the insight related to Deep image prior. Does it mean CR can introduce noise/sharpness that might be be unwanted (e.g., CR results in Figure 3), but at the same time also make the inversed image look sharper?
> >
> > - Does using predicted noise from the previous step (Eq. 12) make the gradient less table?

---

> > > ### Author Response · Authors · 2024-11-25
> > > **Responses to Reviewer tzUN**
> > >
> > > Dear Reviewer tzUN，
> > > ﻿
> > >
> > > Thanks for your feedback.
> > >
> > >
> > > ---
> > >
> > > *Regarding your first question:* We believe that the CR itself should not introduce noise/sharpness. Instead, the observed issue may stem from discretization errors in the differential equations. These errors prevent us from obtaining adjacent $\mu_t$ values with complete accuracy. Over time, the accumulation of such errors might lead to noise/sharpness in the results. The stochasticity introduced by hybrid noise could potentially contract these errors during the inversion process.
> > >
> > > ---
> > >
> > >
> > > *Regarding your second question:* We conducted an ablation study on inpainting using the FFHQ dataset. The results are summarized in the table below. Using the prediction noise from the previous step produces results similar in quality to the original, it reduces computational time by lowering the number of function evaluations (NFE).
> > >
> > > |                      | LPIPS ↓ | FID ↓  | Time (s/img) ↓ |
> > > |----------------------|----------|--------|----------------|
> > > | w/o prev. noise| 0.0807 | 15.39  | 9              |
> > > | w/ prev. noise| 0.0816     | 15.42 | 5              |

---

> > > > ### Comment · Reviewer_tzUN · 2024-11-25
> > > >
> > > > Thanks for the analysis. I tend to keep my score at the moment.

---

### Official Review · Reviewer_Yt8A · 2024-10-31

**Soundness:** 1
**Presentation:** 2
**Contribution:** 1
**Rating:** 6
**Confidence:** 5

**Summary:**

This paper builds on the concept of denoising regularization (DR) for solving inverse problems. Inspired by recent advancements like Red-Diff, which utilizes denoising score matching as a regularization term, the authors introduce a new approach termed consistency regularization (CR). CR adopts the central idea of consistency distillation (CD), which improves self-consistency by minimizing the difference between outputs at two consecutive reverse sampling points. However, the authors note that a direct application of CD can lead to oversaturation and artifacts due to insufficient randomness in the noise sampling process within CD. To address this, they propose a hybrid regularization (HR) framework, which incorporates hybrid noise modeling from the DDIM formulation. Experimental results suggest that HR achieves state-of-the-art performance in various inverse problem settings.

**Strengths:**

The use of a hybrid noise model for score distillation is a notable advance compared to DR and CR.

**Weaknesses:**

Despite its contributions, the paper has several notable limitations. First, to substantiate HR's advantages over DR, a comparative ablation study incorporating the hybrid noise model into the DR framework should have been conducted. The reviewer suspects that HR's benefits stem from mitigating DR's mode-collapse behavior due to the hybrid noise model rather than the use of CR, a point the current manuscript and experiments have not sufficiently addressed.

Second, although the hybrid noise model may appear novel within the Red-Diff framework, the final formulation lacks originality. Specifically, the hybrid noise model and gradient step in equation (11) represent a simplified, one-step version of the decomposed diffusion sampling (DDS) framework [1]. In DDS, the denoised image, obtained through Tweedie’s formula, is updated using a multi-step conjugate gradient (CG) approach, after which hybrid noise is introduced within the DDIM framework. The authors appear unaware of DDS, as there are no comparisons against it.

[1] Chung, H., Lee, S., & Ye, J. C. Decomposed Diffusion Sampler for Accelerating Large-Scale Inverse Problems. In The Twelfth International Conference on Learning Representations. 2024

Third, the reviewer finds the authors' assertion about the growing interest in DR for inverse problems unconvincing. Posterior sampling remains predominant in achieving state-of-the-art results in inverse problems. Moreover, contrary to the authors' claims, recent approaches such as DDS bypass Jacobian computation by leveraging piecewise linear manifold structures. Therefore, this assertion appears unsupported.

Finally, the literature review and experimental comparisons with the latest approaches such as DDS,  are insufficient, lacking several significant recent works.

**Questions:**

- Ablation study: conduct ablation study where the DR formulation combined with hybrid noise model
- Difference: please state the difference between the one-step DDS formulation and the proposed hybrid noise model based HR approaches. Aside from the use of the adjacent sanpling time, what is the difference?
- Comparative study with DDS formulation

---

> ### Author Response · Authors · 2024-11-22
> **Responses to Reviewer Yt8A**
>
> Dear Reviewer Yt8A,
>
> Thank you for your time and feedback on our manuscript, We have carefully addressed your questions and concerns point by point below.
> ***
> **Q: Ablation study for DR formulation combined with a hybrid noise model.**
>
> **A:** Following your suggestion, we combine DR's formulation with the DDIM framework of hybrid noise to perform ablation experiments. The results are shown as follows:
>
> **ImageNet-Inpainting**
> ﻿| Method       | LPIPS ↓   | FID ↓    | CA ↑    |
> |--------------|-----------|----------|---------|
> | DR           | 0.117    | 24.6     | 69.5    |
> | DR+DDIM      | 0.113   | 23.5  | 70.4    |
> | HRDIS (Ours) | **0.096**    | **20.1**   | **71.3**    |
>
> **ImageNet-Super Resolution**
> | Method       | LPIPS ↓   | FID ↓    | CA ↑    |
> |--------------|-----------|----------|---------|
> | DR           | 0.249     | 44.2     | 65.8    |
> | DR+DDIM      | 0.141    | 35.6   | 65.7    |
> | HRDIS (Ours) | **0.138**    | **33.0**   | **68.8**    |
>
> **FFHQ-Nonlinear Deblurring**
> | Method       | LPIPS ↓   | FID ↓    |
> |--------------|-----------|----------|
> | DR           | 0.329   | 80.1     |
> | DR+DDIM      | 0.295    | 73.7     |
> | HRDIS (Ours) | **0.236**     | **52.4**     |
>
> We find that DR+DDIM improves the RED-diff. It actually aligns with the conclusions of Sec.3.1 in the manuscript. The random perturbation of the RED-diff creates uncertainty, but the DDIM solver may alleviate this issue.
> ﻿
> However, DR+DDIM still underperforms compared to HRDIS. HRDIS provides smooth interpolation of DR with CR and is a more flexible framework capable of forming synergies between the two. We have added this discussion on **Page 21** of the revised manuscript and **Figure 15 (Page 21)** illustrates the qualitative comparison.
>
> ***
> **Q: Difference with one-step DDS and comparison with DDS.**
>
> **A:** **DDS [1]:** DDS can be considered as approximate posterior sampling, which is realized by gradually guiding the unconditional DDIM sampling. DDS involves three steps:
>
> Step1: predict $x_{0|t}$ with $x_t$. Step2: solve $\min_{{x}^\prime_{0|t}}1/2 \|\|y-A{x}^\prime_{0|t} \|\|^2 + \gamma/2\|\|{x}^\prime_{0|t}-x _ {0|t}\|\|^2$ (**Eq.1**) with conjugate gradient (CG). Step3: calculate $x_{t-1}$ with DDIM.
>
> DDS iteratively makes modifications for sampling trajectories $x_t, x_{t-1},...,x_0$ by **Eq.1**.
>
> **HRDIS:** Our HRDIS is rooted in the RED-diff framework and treats sampling as a stochastic optimization. A key distinction is that HRDIS maintains an additional variable $\mu$ (initialized as $A^Ty$) throughout the optimization process, which is not present in DDS and has a distinct role from ${x}^\prime_{0|t}$. This variable is updated using gradients $\nabla_{\mu}L_{HR}$ with an off-the-shelf optimizer (Adam).
>
> These fundamental differences result in distinct sampling behaviors.  We show the evolution of the sampling process on **Figure 16 (Page 22)** of the revised manuscript. DDS exhibits a gradual change from **noise to the reconstruction**. HRDIS exhibits a **evolution bridging the degraded image and the reconstruction**.
>
> **Comparison with DDS:** Since DDS (https://github.com/HJ-harry/DDS/) only considers linear measurement. We conduct experiments on inpainting, super-resolution (SR) and compressed sensing (CS). The results are summarized below:
>
> |Task|        | LPIPS ↓ (ImageNet) | FID ↓ (ImageNet) | CA ↑ (ImageNet) | LPIPS ↓ (FFHQ) | FID ↓ (FFHQ) |
> |----|------------|--------------------|------------------|------------|----------------|--------------|
> || DDS  | 0.130 | 27.6 | 67.9  | 0.109 | 22.1 |
> | Inpainting| HRDIS (Ours) |  **0.096**   |  **20.1**   |  **71.3** |    **0.082**  |  **15.4** |
> || DDS          | 0.198    | 41.6 | 62.8      | 0.213   | 41.8           |
> |SR| HRDIS (Ours) |  **0.137**   | **33.0** | **68.8**      | 0.213    | **41.0**       |
> || DDS          | 0.270     | 58.4 | 55.1     | 0.256    | 85.9           |
> |CS| HRDIS (Ours) | **0.059**  | **22.0**   | **72.3**     | **0.088**   | **35.3**          |
>
> HRDIS outperforms DDS on most tasks. **Figure 17 (Page 22)**  of the manuscript shows some qualitative results. We observe that DDS may produce samples that are inconsistent with context.
>
> ***
> **Q: Assertion about inverse problem solver.**
>
> **A:** We fully agree with the reviewer's statement. Approximate posterior sampling is currently the mainstay of diffusion model-based solvers (in terms of image quality and generality). The work in our paper demonstrates for the first time that RED-diff-like stochastic optimization methods can compete with posterior sampling, or even better, in both of these aspects. We expect that our work can bring new algorithmic opportunities into this field.
>
> ***
> [1] *Chung et al. Decomposed Diffusion Sampler for Accelerating Large-Scale Inverse Problems. ICLR 2024*

---

> ### Comment · Reviewer_Yt8A · 2024-11-24
>
> Thank you for addressing some of my concerns by providing an ablation study.  Therefore, I will increase the rating if the following remaining concerns are addressed.
>
> - Given that DR + DDIM improves performance, the remaining difference and advantage of the proposed method stem from the use of a different time step for the distillation ($t + \epsilon$). However, the authors have not provided any ablation study or theoretical analysis to confirm the significance of this step. To address this, the authors should conduct an ablation study by varying the $\epsilon$ value, as DR + DDIM can be considered a special case of the proposed method.
>
> - Step 2 of DDS is equivalent to Equation (11), as the last regularization term can be represented by Equation (12), except for the difference in the time step. The DDS notation $x_{0t}$  in the authors' response is, in fact, equivalent to
>  $\hat \mu_{0|t}$ in the authors' notation, and DDS cost function is also minimized with respect to $\mu$.  Therefore, the visualization in Fig. 16 is not a fair comparison. The noisy sample is displayed for DDS, while Tweedie's posterior mean is presented for the proposed method.   This inconsistency may misrepresents the similarity of DDS relative to the proposed method.

---

> ### Author Response · Authors · 2024-11-25
> **Responses to Reviewer Yt8A**
>
> Dear  Reviewer Yt8A,
>
> Thanks for your insightful feedback and suggestions.
>
> ---
>
> *Regarding your first concern:* We conducted an ablation study on $\varepsilon$ in the CR item, focusing on inpainting and super-resolution tasks using the FFHQ dataset.  The quantitative results are presented in **Figure 20 (Page 25)** of the revised manuscript. Assuming the diffusion model operates over the time interval $[0,1]$, we found that $\varepsilon$ performs well when it is around the order of $1e-2$.
>
> * When $\varepsilon$ is too small ($1e-3$), the effect of CR diminishes, leading to blurred results.
>
> * When $\varepsilon$ is too large ($1e-1$), the resulting image quality degrades due to the discretization error.
>
> Additionally, **Figure 21 (Page 25)** includes qualitative visualizations that illustrate these effects, providing a clearer understanding of how $\varepsilon$ influences the final output.
>
> ---
>
> *Regarding your second concern:*  As per your suggestion, we visualized the evolution of Tweedie's mean in DDS and included the modified visualization in **Figure 16 (Page 22)** of the revised manuscript. Our findings indicate that DDS’s sampling process still differs fundamentally from our proposed method. Specifically, the evolution in DDS does not exhibit the same smooth transition from degraded to reconstructed images as seen in our approach. This difference highlights the unique characteristics and advantages of our method.

---

> > ### Comment · Reviewer_Yt8A · 2024-11-25
> >
> > Dear Authors,
> >
> > - Thanks for providing the ablation study for the first comment.  This is quite informative which highlights the major difference of the proposed method. Still it would be great to discuss intuition why the addition of $\epsilon$ improves the performance over DR+DDIM.
> >
> > - In Figure 16, the difference between DDS and HRDIS appears small in the context of super-resolution except for the initialization.   However, it is quite puzzling to see the big difference in image inpainting.  This difference might also originate from the initialization process, particularly if the same temporal step size is applied to both the baseline and HRDIS. To address this, this reviewer recommends an additional ablation study where DDS is initialized similarly to HRDIS. This study could clearly identify the factors driving the improvement in the proposed method, which I suspect are primarily due to two elements: 1) the effect of $\epsilon$, and 2) the initialization strategy.

---

> ### Author Response · Authors · 2024-11-25
> **Responses to Reviewer Yt8A**
>
> Dear Reviewer Yt8A,
> ﻿
>
> Below, we provide further clarifications regarding your concerns:
> ﻿
> ***
> ﻿
> *The intuition for the addition of $\varepsilon$.*  We hypothesize that it is because HRDIS operates with the stochastic optimizer Adam. The original  $\varepsilon$ scheduling in DDIM may not directly align with our framework.
> ﻿
> ***
> *Differences in sampling behavior.* We would like to address what seems to be a little misunderstanding about our approach and its distinction from DDS. To clarify, HRDIS  always maintains an additional optimization variable, $\mu$, throughout the optimization process. In contrast, DDS  discards $x_{0|t}^{\prime}$  after computing $x_{t-1}$ at next step. This fundamental difference leads to one of the key implications:
> ﻿
> 1) **Ancestral Sampling in DDS:**  DDS must always start from Gaussian noise due to its reliance on ancestral sampling of diffusion model, limiting its flexibility.
> ﻿
> 2) **Flexible Initialization in HRDIS:** HRDIS supports flexible initialization of $\mu$, enabling it to potentially achieve better results.
> ﻿
> As far as we are aware, DDS does not offer an equivalent capability for flexible initialization similar to our HRDIS. However, if we have misunderstood or overlooked any aspects of DDS's behavior, we would greatly appreciate further clarification.

---

> > ### Comment · Reviewer_Yt8A · 2024-11-25
> >
> > Dear Authors,
> >
> > Thank you for the prompt response. However, I believe some points require further clarification:
> >
> > - **The use of stochastic Adam optimization**:  While I appreciate your intuition, it remains unclear why the use of the Adam optimizer specifically relates to the $\epsilon$ offset (i.e., $t+\epsilon$). Even with DR and DDS with $\epsilon=0$, the stochastic Adam optimizer can still be employed as an optimizer. Could you elaborate on this connection?
> >
> > - **Initialization**: It seems there may have been some misunderstanding regarding my recommendation for the ablation study. Incorporating a similar initialization is relatively straightforward, as DDS can begin with CG optimization around the same initialization, followed by the addition of DDIM noise. My suggestion for the ablation study aims to identify potential limitations in the existing DDS formulation by examining: 1) Gaussian initialization, 2) no perturbed time step (i.e. $\epsilon=0$).   This ablation study could confirm the advantages of HDRID by explicitly highlighting the sources of improvement rather than relying on broad, high-level justifications.
> >
> > - **The use of $\mu$**: The reviewer  has difficulty in understanding the advantage of maintaining the additional variable $\mu$ throughout the sampling process, as it can be readily computed from the noisy sample $x_t$ using Tweedie’s formula. The main distinction appears to be whether to explicitly keep $\mu$ or rely solely on the noisy sample $x_t$, with the differences seeming minor and unrelated to performance. Could you clarify this?

---

> ### Author Response · Authors · 2024-11-26
> **Responses to Reviewer Yt8A**
>
> Dear Reviewer  Yt8A,
>
> Thank you for your comments. Here are the responses we have prepared for you. We would like to clarify your third question first because it is important for understanding our approach.
>
> ***
> **The use of  $\mu$:** To further illustrate the differences, we provide a comparison of graph models **(Figure 22, page 26)** in the revised manuscript.  We refer the reviewer to it.
>
> Taken as a whole, DDS corrects the **ancestral sampling** trajectory by solving a sub-optimization problem at each step. The sampling process of our HRDIS/RED-diff, on the other hand, is considered as **a single stochastic optimization problem**. The optimization result is the reconstructed sample.
>
> Locally, in DDS, $x_{0|t}^{\prime}$ only related to $x_{0|t}$ and $y$, which is easy to followed, since $\min_{{x}^\prime_{0|t}}1/2 \|\|y-A{x}^\prime_{0|t} \|\|^2 + \gamma/2\|\|{x}^\prime_{0|t}-x _ {0|t}\|\|^2$. $x_{0|t}^{\prime}$ has no explicit dependency on $x_{0|t+1}^{\prime}$
>
> We use the superscript $(n)$ to denote the optimization step for brevity. In HRDIS/RED-diff, $u^{(n+1)} = u^{(n)}-\nabla L_{HR/DR}$. It can be seen that $u^{(n+1)}$ and $u^{(n)}$ have explicit dependency. **If we do not maintain the $u^{(n)}$, $u^{(n+1)}$ can not be readily computed only from the noisy sample.** $x_{0|t}^{\prime}$ of DDS is not, it can be computed directly by noisy sample $x_{t}$ as you mentioned.
>
> The above difference leads to a more continuous and smooth transformation of $u$. $ x_{0 | t}^\prime$ does not have this guarantee. **Such difference could gradually increase with the number of iterations, leading to a completely distinct performance of HRDIS and DDS**. Regarding quantitative results, our HRDIS is usually much higher than the DDS. We often observe that DDS fails to harmonize with context $y$.
>
> ***
> **Initialization for the conjugate gradient of DDS:** We can certainly use $A^\top y$ to initialize as a starting point for each sub-optimization problem in DDS. However, it is important to note that $\min_{{x}^\prime_{0|t}}1/2 \|\|y-A{x}^\prime_{0|t} \|\|^2 + \gamma/2\|\|{x}^\prime_{0|t}-x _ {0|t}\|\|^2$ is **a convex problem** (when A is a linear degradation), and its local minimum is the global minimum (such characteristic is significantly different from the HRDIS/RED-diff's stochastic optimization problem). Without considering numerical accuracy, the initial point may have little effect on the solution results. We included an experiment as you mentioned, and we found that DDS still performs worse than HRDIS.
>
> |Task|        | LPIPS ↓ (FFHQ) | FID ↓ (FFHQ) |
> |----|---------|----------------|--------------|
> || DDS with $A^\top y$ for CG| 0.113 | 24.8 |
> | Inpainting| HRDIS (Ours) |     **0.082**  |  **15.4** |
> || DDS   with $A^\top y$ for CG       | 0.227   | 43.1           |
> |SR| HRDIS (Ours) |  **0.213**    | **41.0**       |
> || DDS  with $A^\top y$ for CG        | 0.251    | 83.7           |
> |CS| HRDIS (Ours) | **0.088**   | **35.3**          |
>
>
> ***
> **The use of stochastic Adam optimization:** We believe that $\varepsilon$ in HRDIS should also be re-selected based on empirical results, as Adam has momentum, and direct DDIM scheduling can work but may not be optimal.

---

> > ### Comment · Reviewer_Yt8A · 2024-11-26
> >
> > - Thank you for the clarification and the additional ablation study. The graphical comparison in Figure 22 (page 26) is particularly useful, as it clearly highlights the differences between DDS and HRDIS. In light of this, I will increase my score. Additionally, I strongly recommend moving Figure 22 to the main paper and including an explanation of the differences between DDS and HRDIS.
> >
> > - That said, the argument regarding the convexity of the optimization problem in CG within DDS appears misleading. DDS does not fully solve the optimization problem, as it leads to off-manifold behavior. To the best of my understanding, only a few steps are performed to ensure the solution remains on the manifold.
> >
> > - Moreover, it seems that HRDIS leverages stochastic ADAM to determine the optimal offset value $\varepsilon$, though this is not clearly stated in the current text. If my understanding is correct, I strongly recommend emphasizing this aspect in the main description of the algorithm.
> >
> > - Lastly, I would like to point out that the resulting stochastic ADAM optimization is highly non-convex, which is further complicated by the time-varying nature of the regularization term. As such, convergence may remain an open problem. I strongly recommend that the authors  state this in the limitations section of the paper.

---

> ### Author Response · Authors · 2024-11-26
> **Responses to Reviewer Yt8A**
>
> Dear Reviewer Yt8A,
>
> Thank you for your time and improved score! The constructive discussions with you have significantly strengthened our manuscript.
> ﻿
>
> Based on your suggestions, we will incorporate the comparisons with DDS and the role of $\varepsilon$ in the appendix into the main text. Additionally, as recommended, we will make sure that the discussion of HRDIS convergence is included in the limitations section.
>
> Best,
>
> Authors of Submission 9721

---

### Meta-Review · Area_Chair_DT3V · 2024-12-20

**Metareview:**

This paper introduces a novel hybrid regularization (HR) framework for solving inverse problems using pretrained diffusion models. By combining denoising regularization (DR) and consistency regularization (CR), the proposed HR approach effectively addresses limitations in existing methods, achieving state-of-the-art results across a range of tasks. Its ability to outperform state-of-the-art methods across diverse tasks, combined with its clear articulation and extensive evaluation, make it a strong candidate for acceptance.

**Additional Comments On Reviewer Discussion:**

All reviewers agreed to accept this paper after the detailed discussion.

---

### Decision · Program_Chairs · 2025-01-22

Accept (Poster)